# Rethinking Contrastive Learning for Graph Collaborative Filtering: Limitations and a Simple Remedy

Geon Lee [1]    Sunwoo Kim [1]    Kyungho Kim [1]    Kijung Shin [1]

## Abstract

Graph collaborative filtering (GCF) is a dominant paradigm in recommender systems, where contrastive learning (CL) objectives such as the Sampled Softmax (SSM) loss are widely used for optimization. However, it remains unclear how CL interacts with the prediction mechanism of GCF. By unfolding the prediction mechanism of GCF, we show that the user-item prediction score is computed by aggregating learnable weights over a large number of neighbor pairs formed by the multi-hop neighbors of the user and the item. This analysis suggests that effective optimization critically depends on which neighbor pairs are upweighted during training. Empirically, we find that effective recommendation is achievable by selectively upweighting only a small subset of neighbor pairs whose constituent neighbors are structurally similar to the target user and item, and that the effect of such selective upweighting varies across different neighbor pair types. Based on these findings, we analyze SSM and identify key limitations in its neighbor pair weight update dynamics. To address these limitations, we propose NT-SSM, an effective and principled CL objective that induces type-aware neighbor pair weight update dynamics. Experiments demonstrate consistent performance improvements over SSM across multiple datasets and GCF models.

## 1. Introduction

Graph collaborative filtering (GCF) has emerged as a dominant paradigm in modern recommender systems (He et al., 2020; Pal et al., 2020; Gao et al., 2022). By modeling user–item interactions as a graph, GCF captures collaborative signals through message passing, which is crucial for effective recommendation.

Recently, contrastive learning (CL), typically instantiated via the Sampled Softmax (SSM) loss (Wu et al., 2024), has been widely adopted for optimizing GCF models. Prior work typically explains its effectiveness through the geometric properties of learned embeddings, such as alignment and uniformity (Yu et al., 2022; Lin et al., 2022; Park et al., 2023). While these perspectives provide valuable insights into representation quality, they offer limited understanding of how CL interacts with the prediction mechanism of GCF.

To bridge this gap, we systematically examine the prediction mechanism of GCF. Our analysis reveals that, in the forward pass, GCF computes the user-item prediction score by aggregating learnable weights over a large number of *neighbor pairs* formed between the multi-hop neighbors of the user and those of the item. This suggests that GCF's prediction quality critically depends on which neighbor pairs are upweighted during training.

Based on this understanding, we investigate desirable learning behaviors in GCF. We observe that, in real-world datasets, the number of neighbor pairs involved in the user–item prediction score grows extremely large even within a small number of hops. However, our analysis shows that effective recommendation is achievable by upweighting *only* a small subset of neighbor pairs whose constituent neighbors are structurally similar to the target user and item. Moreover, we find that the impact of selective upweighting varies substantially across neighbor pair types, depending on whether the constituent neighbors are users or items.

Building on these insights, we analyze SSM, a generalization of various CL losses, within GCF to understand how it updates the learnable weights of neighbor pairs in optimization. Our analysis shows that SSM does not upweight all neighbor pairs involved in the positive user-item prediction score. Instead, it implicitly identifies a subset of neighbor pairs whose constituent neighbors are structurally similar to the target item and selectively upweights them. That is, SSM partially encourages the desirable learning behaviors.

However, we find that SSM remains suboptimal in two key aspects. First, during training, SSM determines which neighbor pairs to upweight primarily based only on the

[1]KAIST, Seoul, South Korea. Correspondence to: Geon Lee <geonlee0325@kaist.ac.kr>, Kijung Shin <kijungs@kaist.ac.kr>.

*Proceedings of the 43$^{rd}$ International Conference on Machine Learning*, Seoul, South Korea. PMLR 306, 2026. Copyright 2026 by the author(s).

structural similarity of item-side neighbors to the target item, while underemphasizing user-side structural similarity to the target user, despite the importance of both for effective recommendation. Second, the SSM loss fails to account for the distinct effects of selectively upweighting different neighbor pair types.

To address these limitations, we propose neighbor-type–Aware Sampled Softmax (NT-SSM), an effective and principled CL objective that exhibits desirable learning dynamics in GCF. Specifically, NT-SSM determines which neighbor pair to upweight by jointly accounting for the structural similarity (1) between items and their neighbors as well as (2) between users and their neighbors. In addition, NT-SSM adaptively adjusts which neighbor pairs are selected for upweighting across different neighbor pair types. These design choices enable GCF to selectively and effectively upweight informative neighbor pairs.

Extensive experiments show that NT-SSM consistently improves GCF performance over SSM. Specifically, we empirically demonstrate the importance of adaptively upweighting neighbor pairs according to their types. These results indicate that NT-SSM induces desirable learning dynamics through neighbor pair-type-aware optimization.

Our contributions are summarized as follows:

- **Uncovering GCF Mechanisms.** We analyze GCF's prediction mechanism by unfolding its closed-form expression, revealing how multi-hop neighbor pairs contribute to user–item prediction.
- **Identifying Empirical Insights.** We show that selective, type-aware neighbor pair upweighting constitutes desirable learning dynamics for GCF.
- **Revisiting CL in GCF.** We revisit SSM in GCF and analyze its optimization behavior, identifying key limitations that misalign with these desirable learning dynamics.
- **Designing GCF-Oriented CL.** We propose NT-SSM, a CL objective that induces desirable learning dynamics through adaptive neighbor pair upweighting.

Code and datasets are available at `https://github.com/geon0325/NT-SSM`.

## 2. Related Work

**Graph Collaborative Filtering.** Graph collaborative filtering (GCF) methods model user–item interactions as a bipartite graph and leverage high-order structural signals for recommendation. Early approaches, such as PinSage (Ying et al., 2018) and NGCF (Wang et al., 2019), employ deep graph convolution with nonlinear transformations, while LightGCN (He et al., 2020) simplifies this framework by showing that linear message passing is sufficient for effective and efficient recommendation. Due to effectiveness,

LightGCN has become a representative GCF model and a widely adopted backbone in advanced GCF methods (Yu et al., 2022; Lin et al., 2022; Lee et al., 2024b) and across diverse recommendation scenarios, including bundle (Ma et al., 2022; Kim et al., 2024), multimedia (Zhang et al., 2021; Wei et al., 2023), multi-behavior (Kim et al., 2025), and knowledge graph-based recommendation (Yang et al., 2022; Zou et al., 2022). In this work, we study the core prediction mechanisms of GCF and derive principled insights into their optimization behavior.

**Contrastive Learning in GCF.** Contrastive learning (CL) has become a dominant optimization paradigm in GCF (Yu et al., 2022; Lin et al., 2022; Wu et al., 2021; Wang et al., 2022). Pairwise objectives such as Bayesian Personalized Ranking (BPR) (**?**) can be viewed as early forms of CL objective, and have since been generalized to multi-negative formulations such as Sampled Softmax (SSM) (Wu et al., 2024), which are closely related to InfoNCE-style objectives (Oord et al., 2018). Most existing analyses focus on representation-level effects (e.g., alignment and uniformity) of CL (Wang et al., 2022; Yu et al., 2022). In contrast, we study how CL interacts with the prediction mechanism of GCF by analyzing how contrastive objectives reweight neighbor pairs during optimization. Moreover, while contrastive learning in GCF is often used as an auxiliary objective with augmented views (Wu et al., 2021; Yu et al., 2022; 2024; Lin et al., 2022), we focus on its primary usage for GCF optimization.

## 3. Forward-pass Analysis of GCF

In this section, we analyze the prediction mechanism of GCF. By systematically examining LightGCN, we show that GCF does not measure user–item prediction score via direct user-item embedding comparison. Instead, the forward pass computes the score by aggregating a large number of pairwise embedding interactions between the multi-hop neighbors of the user and those of the item.

### 3.1. Preliminary Analysis of GCF

We adopt LightGCN as our backbone due to its effectiveness and representativeness in GCF. It simplifies GCF by removing non-linearities and captures collaborative signals by linearly propagating embeddings over the user-item bipartite graph $\mathcal{G} = (\mathcal{U} \cup \mathcal{I}, \mathcal{E})$, where $\mathcal{U}$ and $\mathcal{I}$ denote users and items, respectively, and $\mathcal{E}$ denotes observed interactions.

**Standard GCF Formulation.** The embedding propagation rule at the $\ell$-th layer for a user $u$ and an item $i$ is defined as:

$$\mathbf{e}_u^{(\ell+1)} = \sum_{i \in \mathcal{N}_u} \frac{\mathbf{e}_i^{(\ell)}}{\sqrt{|\mathcal{N}_u||\mathcal{N}_i|}}, \quad \mathbf{e}_i^{(\ell+1)} = \sum_{u \in \mathcal{N}_i} \frac{\mathbf{e}_u^{(\ell)}}{\sqrt{|\mathcal{N}_i||\mathcal{N}_u|}},$$

where $\mathcal{N}_u$ and $\mathcal{N}_i$ denote the sets of direct (one-hop) neighbors of user $u$ and item $i$, respectively. The initial embeddings $\mathbf{e}^{(0)}$ are learnable ID embeddings. After propagating through $L$ layers, the final representations are obtained by mean pooling, i.e., $\mathbf{e}_u = \frac{1}{L+1} \sum_{\ell=0}^{L} \mathbf{e}_u^{(\ell)}$.

**Unified Matrix Formulation.** To simplify the analysis, let $\mathbf{R} \in \{0,1\}^{|\mathcal{U}| \times |\mathcal{I}|}$ be the user-item interaction matrix where $\mathbf{R}_{ui} = 1$ if $(u,i) \in \mathcal{E}$ and 0 otherwise. Let $\mathbf{E}_U^{(\ell)} \in \mathbb{R}^{|\mathcal{U}| \times d}$ and $\mathbf{E}_I^{(\ell)} \in \mathbb{R}^{|\mathcal{I}| \times d}$ denote the embedding matrices for users and items at layer $\ell$, respectively. Then, we define the unified embedding matrix $\mathbf{E}^{(\ell)} \in \mathbb{R}^{(|\mathcal{U}|+|\mathcal{I}|) \times d}$ and the adjacency matrix $\mathbf{A} \in \{0,1\}^{(|\mathcal{U}|+|\mathcal{I}|) \times (|\mathcal{U}|+|\mathcal{I}|)}$ as:

$$\mathbf{E}^{(\ell)} = \begin{bmatrix} \mathbf{E}_U^{(\ell)} \\ \mathbf{E}_I^{(\ell)} \end{bmatrix}, \qquad \mathbf{A} = \begin{bmatrix} \mathbf{0} & \mathbf{R} \\ \mathbf{R}^\top & \mathbf{0} \end{bmatrix}.$$

Let $\widetilde{\mathbf{A}} = \mathbf{D}^{-\frac{1}{2}} \mathbf{A} \mathbf{D}^{-\frac{1}{2}}$ denote the symmetrically normalized adjacency matrix, where $\mathbf{D} = \text{diag}(\mathbf{A}\mathbf{1})$. Then, the propagation rule can be written as a matrix power iteration:

$$\mathbf{E}^{(k)} = \widetilde{\mathbf{A}} \mathbf{E}^{(k-1)} \quad \Rightarrow \quad \mathbf{E}^{(k)} = \widetilde{\mathbf{A}}^k \mathbf{E}^{(0)}.$$

Consequently, the final embedding matrix $\mathbf{E}$ is a fixed linear transformation of the initial embeddings:

$$\mathbf{E} = \frac{1}{L+1} \sum_{\ell=0}^{L} \mathbf{E}^{(\ell)} = \left( \frac{1}{L+1} \sum_{\ell=0}^{L} \widetilde{\mathbf{A}}^\ell \right) \mathbf{E}^{(0)} = \widetilde{\mathbf{S}} \mathbf{E}^{(0)},$$

where $\widetilde{\mathbf{S}} \in \mathbb{R}^{(|\mathcal{U}|+|\mathcal{I}|) \times (|\mathcal{U}|+|\mathcal{I}|)}$ is the structural similarity matrix that contains no learnable parameters. An entry $\widetilde{\mathbf{S}}_{uv}$ quantifies the structural weight between nodes $u$ and $v$ aggregated over $L$ hops.

**Multi-hop Neighbors.** For any node $u \in \mathcal{U} \cup \mathcal{I}$, we define its multi-hop neighbors $\widetilde{\mathcal{N}}_u = \{v \mid \widetilde{\mathbf{S}}_{uv} \neq 0\}$ as the set of all nodes reachable within $L$ steps. This set can be decomposed into disjoint user- and item-type neighbors, $\widetilde{\mathcal{N}}_u = \widetilde{\mathcal{N}}_u^{(U)} \cup \widetilde{\mathcal{N}}_u^{(I)}$, where $\widetilde{\mathcal{N}}_u^{(U)}$ contains user-type neighbors and $\widetilde{\mathcal{N}}_u^{(I)}$ contains item-type neighbors, with $\widetilde{\mathcal{N}}_u^{(U)} \cap \widetilde{\mathcal{N}}_u^{(I)} = \varnothing$. Then, the final embedding $\mathbf{e}_u$ can be decomposed into contributions from each neighbor-type:

$$\mathbf{e}_u = \sum_{v \in \widetilde{\mathcal{N}}_u} \widetilde{\mathbf{S}}_{uv} \mathbf{e}_v^{(0)} = \sum_{v \in \widetilde{\mathcal{N}}_u^{(U)}} \widetilde{\mathbf{S}}_{uv} \mathbf{e}_v^{(0)} + \sum_{v' \in \widetilde{\mathcal{N}}_u^{(I)}} \widetilde{\mathbf{S}}_{uv'} \mathbf{e}_{v'}^{(0)}.$$

### 3.2. Prediction Mechanism of GCF

We now analyze how the structural aggregation defined above translates into the final prediction.

**Prediction Score via Neighbor Pair Aggregation.** The final prediction score $\hat{r}_{ui}$ is typically computed as the inner

product of the final user and item embeddings. By substituting $\mathbf{E} = \widetilde{\mathbf{S}} \mathbf{E}^{(0)}$, we can expand this score as:

$$\hat{r}_{ui} = \left( \sum_{v \in \widetilde{\mathcal{N}}_u} \widetilde{\mathbf{S}}_{uv} \mathbf{e}_v^{(0)} \right)^\top \left( \sum_{v' \in \widetilde{\mathcal{N}}_i} \widetilde{\mathbf{S}}_{iv'} \mathbf{e}_{v'}^{(0)} \right)$$

$$= \sum_{v \in \widetilde{\mathcal{N}}_u} \sum_{v' \in \widetilde{\mathcal{N}}_i} \widetilde{\mathbf{S}}_{uv} \cdot \widetilde{\mathbf{S}}_{iv'} \cdot \underbrace{\left( \mathbf{e}_v^{(0)\top} \mathbf{e}_{v'}^{(0)} \right)}_{\text{Learnable Weight}} \qquad (1)$$

This derivation shows that the prediction score is not based on direct comparison between $u$ and $i$, but instead aggregates learnable weights over *all pairs of neighbors* drawn from their respective multi-hop neighbors.

From this viewpoint, the core learning problem in GCF is to determine how neighbor pairs should be desirably weighted. Equivalently, *GCF learns which neighbor pairs should be upweighted and which should be downweighted.*

**Prediction Score Decomposition.** The final prediction score can be decomposed into four distinct sub-scores based on the neighbor-types of users and items:

$$\hat{r}_{ui} = \hat{r}_{ui}^{(U,U)} + \hat{r}_{ui}^{(I,I)} + \hat{r}_{ui}^{(U,I)} + \hat{r}_{ui}^{(I,U)}, \qquad (2)$$

where $\hat{r}_{ui}^{(t,t')} = \sum_{v \in \widetilde{\mathcal{N}}_u^{(t)}} \sum_{v' \in \widetilde{\mathcal{N}}_i^{(t')}} \widetilde{\mathbf{S}}_{uv} \cdot \widetilde{\mathbf{S}}_{iv'} \cdot \left( \mathbf{e}_v^{(0)\top} \mathbf{e}_{v'}^{(0)} \right),$

for $t, t' \in \{U, I\}$. This reveals that GCF predicts user-item prediction score by collectively aggregating similarities from four distinct types of neighbor pairs: User-User (UU), Item-Item (II), User-Item (UI), and Item-User (IU).

## 4. Learning Dynamics Analysis of GCF

Recall that we analyzed the user–item prediction score in a representative graph collaborative filtering (GCF) model, LightGCN. Specifically, we unfolded its closed-form expression and found that node pairs between multi-hop neighbors of the target user and target item also contribute to the score. In this section, we examine the learning dynamics of the learnable components in the unfolded expression (Eq. (1)).

### 4.1. Implication of Learning Dynamics in GCF

We examine how standard training in GCF influences the learnable components in Eq. (1).

**Settings and Implication.** Consider an observed user–item interaction between user $u$ and item $i$, which often serves as a positive training sample in GCF. For such a user–item pair, model parameters are optimized to maximize the pair's prediction score (Eq. (1)). In Eq. (1), the learnable components are the inner-products between (1) the initial embeddings

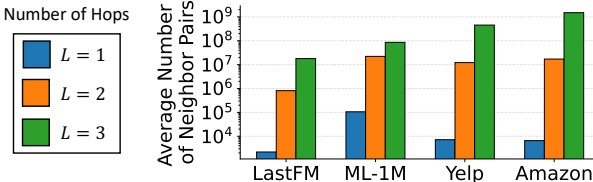

*Figure 1.* The average number of neighbor pairs involved in computing user–item relevance (Eq. (1)) becomes substantially large as the hop increases across different datasets.

of the $L$-hop neighbors of user $u$, and (2) the initial embeddings of the $L$-hop neighbors of item $i$, which we refer to as *learnable weights*. Since all entries in the structural similarity matrix $\widetilde{\mathbf{S}}$ are non-negative, upweighting these terms directly leads to a higher prediction score in Eq. (1).

**Empirical Findings.** However, as shown in Figure 1, the number of learnable weights in Eq. (1) (i.e., the number of neighbor pairs) is extremely large, even within a small number of hops. For example, on Amazon-Book, the number of neighbor pairs involved in computing a prediction score reaches billions on average at three hops, while on MovieLens, more than 90% of all possible neighbor pairs are already included. Since many user–user, item–item, and user–item pairs are *irrelevant* (e.g., users with different purchase patterns and items with different purchased users), this observation suggests that upweighting nearly all possible node pairs is intuitively undesirable.

### 4.2. Desirable Learning Dynamics

Based on the findings in Section 4.1, we study learning dynamics that better align with our intuition and empirically demonstrate their effectiveness in recommendation. We then discuss the implications of our investigation for analyzing and improving the learning process of GCF models.

**Hypothesis.** Consider a neighbor pair $(v, v')$ such that $v \in \widetilde{\mathcal{N}}_u$ and $v' \in \widetilde{\mathcal{N}}_i$. When $v$ and $v'$ are structurally similar to $u$ and $i$, respectively, we expect this pair to contribute more effectively to the prediction score $\hat{r}_{ui}$. Based on this intuition, rather than increasing learnable weights for all pairs involved in the prediction score $r_{ui}$ (Eq. (1)), we hypothesize that it is more effective to focus on neighbors that are highly structurally similar to $u$ and $i$.

**Empirical setup.** To validate our hypothesis, we conduct an empirical study that controls which neighbor pairs contribute to the prediction score based on their structural similarity. Specifically, when computing the prediction score between user $u$ and item $i$, we use only those $L$-hop neighbors of user $u$ that exhibit high structural similarity to $u$, and apply the same criterion to the neighbors of item $i$, rather than using all $L$-hop neighbors. For node $x \in \{u, i\}$, we denote its neighbor set with high structural similarity

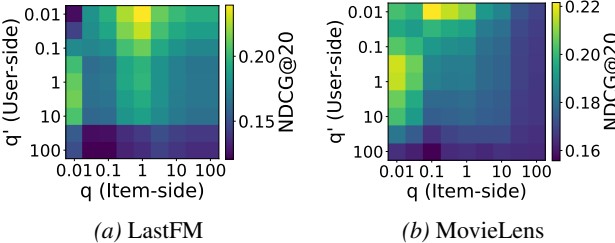

*(a)* LastFM   *(b)* MovieLens

*Figure 2.* Recommendation performance (NDCG@20) using the heuristic prediction score (Eq. (4)) under different user and item neighbor retention ratios $q$ and $q'$. Increasing the weights of neighbor pairs whose constituent neighbors are structurally similar to the respective user and item yields the best performance.

as $\widetilde{\mathcal{N}}_x^{\langle q \rangle}$, where $q$ is the *retention ratio* used to select the top-$q\%$ $L$-hop neighbors with highest structural similarity:

$$\widetilde{\mathcal{N}}_x^{\langle q \rangle} = \left\{ v \in \widetilde{\mathcal{N}}_x \;\middle|\; \widetilde{\mathbf{S}}_{xv} \geq \text{top}_{q\%}\big(\widetilde{\mathbf{S}}_{\cdot,v}\big) \right\}.$$

Then, for a neighbor pair $(v, v')$, where $v \in \widetilde{\mathcal{N}}_u^{\langle q \rangle}$ and $v' \in \widetilde{\mathcal{N}}_i^{\langle q' \rangle}$, we count their co-occurrences within the retained neighbor sets across observed user-item interactions:

$$\omega_{q,q'}(v, v') = \sum_{(u',i') \in \mathcal{E}} \mathbb{I}\big[(v \in \widetilde{\mathcal{N}}_{u'}^{\langle q \rangle}) \wedge (v' \in \widetilde{\mathcal{N}}_{i'}^{\langle q' \rangle})\big], \quad (3)$$

where distinct retention ratios $q$ and $q'$ can be applied to the neighbors of users and items, respectively. We replace each node pair's learnable weight in Eq. (1) with the corresponding node pair's co-occurrence count (Eq. (3)) and compute the heuristic prediction score as:

$$\hat{r}_{ui}^{\langle q,q' \rangle} = \sum_{v \in \widetilde{\mathcal{N}}_u} \sum_{v' \in \widetilde{\mathcal{N}}_i} \widetilde{\mathbf{S}}_{uv} \cdot \widetilde{\mathbf{S}}_{iv'} \cdot \omega_{q,q'}(v, v'). \quad (4)$$

This heuristic prediction score serves as a proxy for analyzing how selectively using neighbor pairs affects recommendation performance.

**Empirical Observations.** We evaluate the heuristic prediction score (Eq. (4)) under varying neighbor retention ratios $q$ and $q'$, and make the following observations that provide insights into desirable learning dynamics in GCF.

**Observation 1.** *Using only a small subset of neighbor pairs with high structural similarity leads to improved recommendation performance.*

As shown in Figure 2, accounting for co-occurrences among only a small fraction of structurally similar neighbors, and thus selectively upweighting the induced neighbor pairs, for *both* users and items, outperforms using all neighbors. For example, on MovieLens, retaining $0.01\%$ of user neighbors ($q = 0.01$) and $0.1\%$ of item neighbors ($q' = 0.1$) achieves the best performance, resulting in a 35.17% improvement in NDCG@20 over using all neighbors ($q = q' = 100$). These results provide empirical evidence that *selectively*

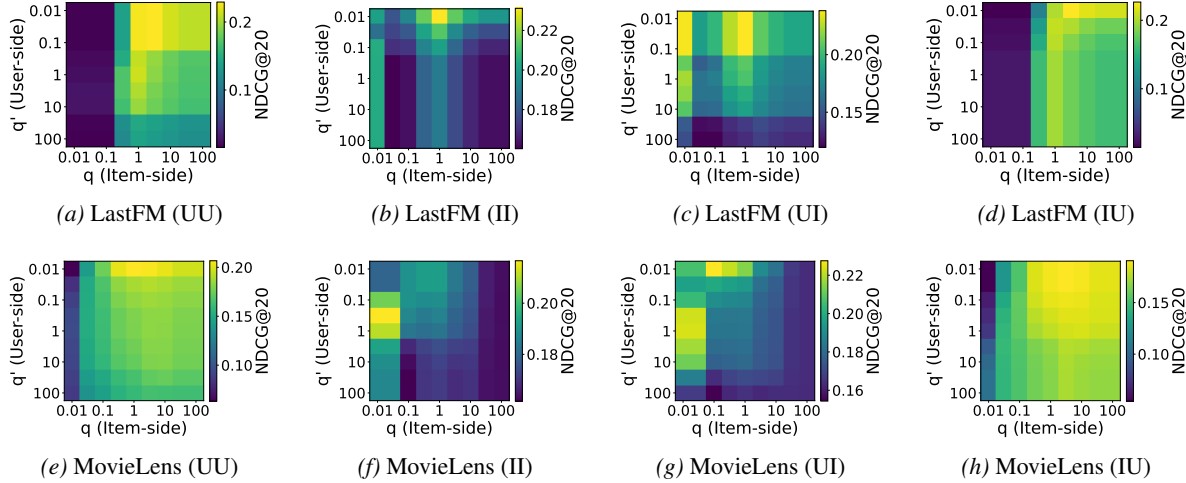

*(a) LastFM (UU)*     *(b) LastFM (II)*     *(c) LastFM (UI)*     *(d) LastFM (IU)*

*(e) MovieLens (UU)*     *(f) MovieLens (II)*     *(g) MovieLens (UI)*     *(h) MovieLens (IU)*

*Figure 3.* Recommendation performance (NDCG@20) of individual neighbor pair types (UU, II, UI, and IU) under varying neighbor retention ratios $q$ and $q'$. Different neighbor pair types exhibit distinct performance patterns and optimal retention ratios across datasets, indicating that the effect of selectively updating neighbor pairs is highly type-dependent.

*upweighting neighbor pairs whose constituent neighbors are structurally similar to the respective user and item* leads to more desirable learning dynamics in GCF.

**Observation 2.** *The effect of selectively upweighting neighbor pairs varies across different neighbor pair types.*

As discussed in Section 3, neighbor pairs can be categorized into different types (i.e., UU, II, UI, and IU) based on the types of the involved nodes (i.e., users or items). Accordingly, we decompose Eq. (4) according to neighbor pair types, as formalized in Eq. (2), and evaluate the performance of each type separately. As shown in Figure 3, different neighbor pair types exhibit distinct performance trends under varying user and item neighbor retention ratios. In particular, optimal retention ratios vary across neighbor-types and datasets. These results indicate that the effectiveness of upweighting neighbor pairs is highly dependent on the neighbor pair type, implying that learning dynamics in GCF can benefit from *adaptively upweighting neighbor pairs based on their types.*

## 5. Learning Dynamics of CL in GCF

We now analyze the learning dynamics of GCF under contrastive learning (CL) and examine how these dynamics align with the desirable learning behaviors discussed in Section 4. We focus on the Sampled SoftMax (SSM) loss (Wu et al., 2024) as the contrastive loss, which is closely related to InfoNCE loss (Oord et al., 2018) and generalizes Bayesian Personalized Ranking loss (**?**).

### 5.1. How SSM Updates Neighbor Pair Weights

We first review the SSM loss and analyze its gradient.

**SSM Loss.** For a user $u$, a positive item $i$, and a sampled set of negative items $\mathcal{B}_u$, the SSM loss is defined as:

$$\mathcal{L}(i; u) = -\log \frac{\exp\big(s(u,i)/\tau\big)}{\exp\big(s(u,i)/\tau\big) + \sum_{j \in \mathcal{B}_u} \exp\big(s(u,j)/\tau\big)},$$

where $s(u,i)$ denotes similarity between $u$ and $i$ (e.g., inner product or cosine similarity), $\mathcal{B}_u$ is the set of negative items sampled for user $u$, and $\tau$ is a temperature hyperparameter that controls the sharpness of the softmax distribution. The overall SSM loss is $\frac{1}{|\mathcal{E}|} \sum_{(u,i)\in\mathcal{E}} \mathcal{L}(i; u)$.

**Weight Update Dynamics under SSM.** To examine how the SSM loss governs similarity update dynamics over neighbor pairs in GCF, we analyze the gradient of the loss with respect to the learnable weight term $\mathbf{e}_v^{(0)\top}\mathbf{e}_{v'}^{(0)}$ associated with a neighbor pair $(v, v')$:

$$\frac{\partial \mathcal{L}(i; u)}{\partial(\mathbf{e}_v^{(0)\top}\mathbf{e}_{v'}^{(0)})} = \frac{\widetilde{\mathbf{S}}_{uv}}{\tau}\left(\mathbb{E}_{x\sim\pi_u}\big[\widetilde{\mathbf{S}}_{xv'}\big] - \widetilde{\mathbf{S}}_{iv'}\right), \quad (5)$$

where $\pi_u$ denotes the model-induced distribution over the items, which depends on the negative sampling strategy (see Appendix A for detailed derivations).[1]

**Alignment Between SSM- and Desirable Dynamics.** Under gradient descent, if Eq. (5) is positive, the pair $(v, v')$ is downweighted; if it is negative, the pair is upweighted. Therefore, given that all elements in $\widetilde{\mathbf{S}}$ are non-negative, the sign of $\mathbb{E}_x[\widetilde{\mathbf{S}}_{xv'}] - \widetilde{\mathbf{S}}_{iv'}$ decides how the learnable weight of the neighbor pair is updated. Intuitively, this result indicates that when a structural similarity between node $v'$ and item $i$

---

[1]For simplicity, we use the inner product, i.e., $s(u,i) = \mathbf{e}_u^\top\mathbf{e}_i$, in the analysis. We can use cosine similarity following the standard SSM setting; the extension is detailed in Appendix D.

($\widetilde{\mathbf{S}}_{iv'}$) is greater than the expected structural similarity between node $v'$ and an arbitrary node $x$ ($\mathbb{E}_x[\widetilde{\mathbf{S}}_{xv'}]$), the pair $(v, v')$ is upweighted.

Surprisingly, this result aligns with the desirable learning dynamics discussed in Section 4. Specifically, among the neighbors of item $i$ (i.e., $v' \in \widetilde{\mathcal{N}}_i$), the desirable dynamics suggest upweighting neighbor pairs that include nodes that are structurally similar to $i$ (Observation 1). Consistently, the SSM loss follows a similar principle: it upweights neighbor pairs involving $v'$ only when the structural similarity between $v'$ and $i$ exceeds the random-baseline expectation. Therefore, both approaches indicate that neighbor pairs involving nodes that are structurally similar to the target item $i$ should be primarily upweighted.

### 5.2. Limitations of Learning Dynamics Induced by SSM

While the SSM loss aligns with the desirable dynamics, our gradient analysis reveals several areas for improvement.

**Limitation 1.** *The learning dynamics of neighbor pair weights are centered primarily on items' neighbors.*

As shown by the gradient analysis, whether a neighbor pair is upweighted under SSM depends only on the structural similarity of the item-side neighbor to the target item (see Eq. (5)), while the structural similarity of the user-side neighbor to the target user is not taken into account. However, Observation 1 in Section 4 shows that effective recommendation requires jointly considering structurally similar neighbors for both users and items. Therefore, the current SSM loss formulation, which centers on item-side structural similarity, can be suboptimal.

**Limitation 2.** *The learning dynamics apply the same upweighting and downweighting criterion across different types of neighbor pairs.*

The gradient of SSM reveals that it applies the shared rule to upweight all neighbor pair types, including UU, II, UI, and IU. However, Observation 2 in Section 4 shows that different neighbor pair types exhibit distinct optimal behaviors under selective upweighting. By applying a uniform rule to upweight all neighbor pair types, SSM overlooks these type-dependent effects, which can be suboptimal.

## 6. Proposed CL Objective for GCF

In this section, we propose neighbor-type–Aware Sampled Softmax (NT-SSM), a simple yet principled contrastive learning objective that induces desirable learning dynamics in GCF and effectively addresses the limitations of SSM (Figure 4). We further extend BPR to its neighbor-type–aware variant, NT-BPR, as detailed in Appendix C.

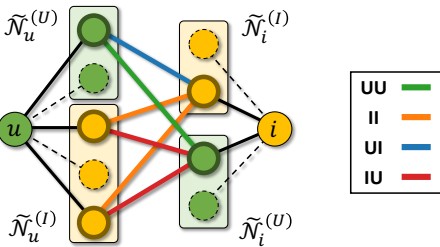

*Figure 4.* For a target user–item interaction $(u, i)$, NT-SSM decomposes neighbor pairs into four types: user–user (UU), item–item (II), user–item (UI), and item–user (IU). NT-SSM selectively upweights neighbor pairs with structurally similar neighbors (those with bold borders) and adaptively controls their update dynamics using neighbor pair type-specific coefficients.

### 6.1. Neighbor-Type–Aware Sampled Softmax

We introduce NT-SSM, a CL objective tailored to GCF that more flexibly and adaptively controls the dynamics of neighbor pair upweighting.

Given an observed (positive) interaction $(u, i) \in \mathcal{E}$, NT-SSM optimizes a *bidirectional* contrastive objective that accounts for predictions from both the user perspective and the item perspective:

$$\mathcal{L}(u, i) = \widetilde{\mathcal{L}}(i; u) + \widetilde{\mathcal{L}}(u; i), \tag{6}$$

where $\widetilde{\mathcal{L}}(i; u)$ contrasts the positive item $i$ against negative items conditioned on user $u$, and $\widetilde{\mathcal{L}}(u; i)$ is defined symmetrically for users conditioned on items.

We define the contrastive loss $\widetilde{\mathcal{L}}(i; u)$ using a neighbor-type–aware similarity function for negative items:

$$\widetilde{\mathcal{L}}(i; u) = -\log \frac{\exp\left(s(u, i)/\tau\right)}{\exp\left(s(u, i)/\tau\right) + \sum_{j \in \mathcal{B}_u} \exp\left(\widetilde{s}(u, j)/\tau\right)},$$

where the negative similarity $\widetilde{s}(u, j)$ is decomposed according to neighbor-types as:

$$\widetilde{s}(u, j) = \alpha_I^{(U)} s_I^{(U)}(u, j) + \alpha_I^{(I)} s_I^{(I)}(u, j),$$

$$s_I^{(t)}(u, j) = \sum_{v \in \widetilde{\mathcal{N}}_u} \sum_{v' \in \widetilde{\mathcal{N}}_j^{(t)}} \widetilde{\mathbf{S}}_{uv} \cdot \widetilde{\mathbf{S}}_{jv'} \cdot (\mathbf{e}_v^{(0)\top} \mathbf{e}_{v'}^{(0)}),$$

where $t \in \{U, I\}$ denotes the neighbor-type. Here, $s_I^{(t)}(u, j)$ is the partial similarity contributed by neighbor pairs involving item $j$'s neighbors of type $t$, i.e., $\widetilde{\mathcal{N}}_j^{(t)}$. The coefficients $\alpha_I^{(t)}$ control how similarity updates are scaled for different neighbor-types, reflecting the observation that desirable learning dynamics are type-dependent..

Setting $\alpha_I^{(U)} = \alpha_I^{(I)} = 1$ and considering only $\widetilde{\mathcal{L}}(i; u)$ reduces NT-SSM to the standard SSM loss. The counterpart $\widetilde{\mathcal{L}}(u; i)$ is defined analogously by decomposing negative users' similarities according to the types of their neighbors.

*Table 1.* Overall recommendation performance of CL objectives, BPR and SSM, and their neighbor-type-aware counterparts, NT-BPR and NT-SSM, respectively, under various GCF models. Both NT-BPR and NT-SSM consistently outperform their corresponding baselines, BPR and SSM, across datasets, demonstrating the effectiveness of neighbor-type-aware weight update learning dynamics.

| Method | Loss | LastFM | | MovieLens | | Yelp | | Amazon-Book | |
|---|---|---|---|---|---|---|---|---|---|
| | | Recall@20 | NDCG@20 | Recall@20 | NDCG@20 | Recall@20 | NDCG@20 | Recall@20 | NDCG@20 |
| LightGCN | BPR | $0.2755_{\pm0.0017}$ | $0.2530_{\pm0.0016}$ | $0.2328_{\pm0.0016}$ | $0.2953_{\pm0.0009}$ | $0.0554_{\pm0.0003}$ | $0.0449_{\pm0.0003}$ | $0.0356_{\pm0.0002}$ | $0.0273_{\pm0.0002}$ |
| | NT-BPR | $\mathbf{0.2897}_{\pm0.0025}$ | $\mathbf{0.2654}_{\pm0.0016}$ | $\mathbf{0.2487}_{\pm0.0008}$ | $\mathbf{0.3154}_{\pm0.0009}$ | $\mathbf{0.0586}_{\pm0.0005}$ | $\mathbf{0.0480}_{\pm0.0004}$ | $\mathbf{0.0393}_{\pm0.0003}$ | $\mathbf{0.0301}_{\pm0.0003}$ |
| | Improv. | 5.15% | 4.90% | 6.83% | 6.81% | 5.78% | 6.90% | 10.39% | 10.26% |
| | SSM | $0.2617_{\pm0.0021}$ | $0.2404_{\pm0.0013}$ | $0.2134_{\pm0.0009}$ | $0.2648_{\pm0.0024}$ | $0.0649_{\pm0.0002}$ | $0.0530_{\pm0.0002}$ | $0.0533_{\pm0.0004}$ | $0.0414_{\pm0.0003}$ |
| | NT-SSM | $\mathbf{0.2953}_{\pm0.0012}$ | $\mathbf{0.2709}_{\pm0.0013}$ | $\mathbf{0.2544}_{\pm0.0005}$ | $\mathbf{0.3216}_{\pm0.0002}$ | $\mathbf{0.0692}_{\pm0.0002}$ | $\mathbf{0.0565}_{\pm0.0001}$ | $\mathbf{0.0542}_{\pm0.0004}$ | $\mathbf{0.0421}_{\pm0.0002}$ |
| | Improv. | 12.84% | 12.69% | 19.21% | 21.45% | 6.63% | 6.60% | 1.69% | 1.69% |
| SimGCL | BPR | $0.2934_{\pm0.0017}$ | $0.2687_{\pm0.0009}$ | $0.2523_{\pm0.0011}$ | $0.3159_{\pm0.0006}$ | $0.0668_{\pm0.0001}$ | $0.0542_{\pm0.0001}$ | $0.0448_{\pm0.0004}$ | $0.0343_{\pm0.0003}$ |
| | NT-BPR | $\mathbf{0.2949}_{\pm0.0011}$ | $\mathbf{0.2697}_{\pm0.0008}$ | $\mathbf{0.2563}_{\pm0.0007}$ | $\mathbf{0.3190}_{\pm0.0014}$ | $\mathbf{0.0674}_{\pm0.0005}$ | $\mathbf{0.0548}_{\pm0.0004}$ | $\mathbf{0.0459}_{\pm0.0003}$ | $\mathbf{0.0352}_{\pm0.0001}$ |
| | Improv. | 0.51% | 0.37% | 1.59% | 0.98% | 0.90% | 1.11% | 2.46% | 2.62% |
| | SSM | $0.2789_{\pm0.0005}$ | $0.2562_{\pm0.0005}$ | $0.2354_{\pm0.0005}$ | $0.2899_{\pm0.0017}$ | $0.0678_{\pm0.0002}$ | $0.0556_{\pm0.0002}$ | $0.0459_{\pm0.0002}$ | $0.0356_{\pm0.0000}$ |
| | NT-SSM | $\mathbf{0.2939}_{\pm0.0013}$ | $\mathbf{0.2726}_{\pm0.0014}$ | $\mathbf{0.2602}_{\pm0.0009}$ | $\mathbf{0.3258}_{\pm0.0006}$ | $\mathbf{0.0680}_{\pm0.0001}$ | $\mathbf{0.0558}_{\pm0.0001}$ | $\mathbf{0.0488}_{\pm0.0003}$ | $\mathbf{0.0383}_{\pm0.0003}$ |
| | Improv. | 5.38% | 6.40% | 10.54% | 12.38% | 0.29% | 0.36% | 6.32% | 7.58% |
| NCL | BPR | $0.2906_{\pm0.0014}$ | $0.2666_{\pm0.0009}$ | $0.2452_{\pm0.0016}$ | $0.3110_{\pm0.0013}$ | $0.0616_{\pm0.0005}$ | $0.0501_{\pm0.0004}$ | $0.0396_{\pm0.0004}$ | $0.0302_{\pm0.0002}$ |
| | NT-BPR | $\mathbf{0.2916}_{\pm0.0018}$ | $\mathbf{0.2695}_{\pm0.0009}$ | $\mathbf{0.2501}_{\pm0.0015}$ | $\mathbf{0.3171}_{\pm0.0016}$ | $\mathbf{0.0626}_{\pm0.0005}$ | $\mathbf{0.0511}_{\pm0.0003}$ | $\mathbf{0.0426}_{\pm0.0002}$ | $\mathbf{0.0330}_{\pm0.0002}$ |
| | Improv. | 0.34% | 1.09% | 2.00% | 1.96% | 1.62% | 2.00% | 7.58% | 9.27% |
| | SSM | $0.2797_{\pm0.0005}$ | $0.2573_{\pm0.0009}$ | $0.2168_{\pm0.0016}$ | $0.2614_{\pm0.0021}$ | $0.0655_{\pm0.0004}$ | $0.0536_{\pm0.0002}$ | $0.0412_{\pm0.0001}$ | $0.0322_{\pm0.0001}$ |
| | NT-SSM | $\mathbf{0.2960}_{\pm0.0021}$ | $\mathbf{0.2708}_{\pm0.0015}$ | $\mathbf{0.2493}_{\pm0.0009}$ | $\mathbf{0.3132}_{\pm0.0006}$ | $\mathbf{0.0669}_{\pm0.0002}$ | $\mathbf{0.0545}_{\pm0.0002}$ | $\mathbf{0.0534}_{\pm0.0002}$ | $\mathbf{0.0414}_{\pm0.0002}$ |
| | Improv. | 5.83% | 5.25% | 14.99% | 19.82% | 2.14% | 1.68% | 29.61% | 28.57% |

## 6.2. How NT-SSM Upweights Neighbor Pairs

We analyze how NT-SSM induces the dynamics of up-weighting neighbor pairs by examining the gradient of $\mathcal{L}(u, i)$ with respect to a neighbor pair weight. The gradient of Eq. (6) (our proposed loss) with respect to the weight of $(v, v')$, where $v \in \widetilde{\mathcal{N}}_u$ and $v' \in \widetilde{\mathcal{N}}_i$, can be written as:

$$\frac{\partial \mathcal{L}(u, i)}{\partial (\mathbf{e}_v^{(0)\top} \mathbf{e}_{v'}^{(0)})} = \frac{\Pi_u \widetilde{\mathbf{S}}_{uv}}{\tau} \left( \alpha_I^{(t_{v'})} \mathbb{E}_{j \sim \hat{\pi}_u} [\widetilde{\mathbf{S}}_{jv'}] - \widetilde{\mathbf{S}}_{iv'} \right)$$
$$+ \frac{\Pi_i \widetilde{\mathbf{S}}_{iv'}}{\tau} \left( \alpha_U^{(t_v)} \mathbb{E}_{k \sim \hat{\pi}_i} [\widetilde{\mathbf{S}}_{kv}] - \widetilde{\mathbf{S}}_{uv} \right),$$

where $t_{v'}$ and $t_v$ denote the neighbor-types of neighbors $v'$ and $v$, respectively. [2] Here, $\Pi_u$ and $\Pi_i$ represent the total probability masses assigned to negative samples, and $\mathbb{E}_{j \sim \hat{\pi}}$ denotes the expectation over the negative distribution. For a detailed derivation, refer to Appendix B.

This derivation shows that the upweighting of neighbor pairs under NT-SSM depends on both user-side and item-side structural similarities, and that the criteria are explicitly modulated by neighbor-type-specific coefficients $\alpha$. That is, a neighbor pair $(v, v')$ is upweighted (i.e., receives a negative gradient update) based on the combined effect of the user-side and item-side contributions:

$$\widetilde{\mathbf{S}}_{iv'} > \alpha_I^{(t_{v'})} \mathbb{E}_{j \sim \hat{\pi}_u} [\widetilde{\mathbf{S}}_{jv'}], \quad \widetilde{\mathbf{S}}_{uv} > \alpha_U^{(t_v)} \mathbb{E}_{k \sim \hat{\pi}_i} [\widetilde{\mathbf{S}}_{kv}].$$

---
[2] That is, $t_v = U$ if $v \in \mathcal{U}$, and $t_v = I$ if $v \in \mathcal{I}$.

Intuitively, when $\alpha < 1$, NT-SSM relaxes the criterion, allowing noisy but potentially useful neighbor pairs to be upweighted. In contrast, when $\alpha > 1$, NT-SSM imposes a stricter criterion, so that only neighbor pairs with sufficiently structurally similar neighbors are upweighted.

## 6.3. How NT-SSM Meets Desirable Learning Dynamics

We now discuss how NT-SSM induces learning dynamics that more closely align with the desirable behavior discussed in Section 4 and addresses the limitations of SSM.

**Property 1.** *NT-SSM induces neighbor pair weight update dynamics accounting for both user- and item-side neighbors.*

Recall that under the SSM gradient (Eq. (5)), whether a neighbor pair is upweighted depends only on the structural similarity of the item-side neighbor to the target item, while the structural similarity between the user-side neighbor to the target user is not taken into account (Limitation 1). In contrast, NT-SSM incorporates update signals from both the user and item sides via its bidirectional objective. As a result, similarity updates under NT-SSM reflect structural similarity on both sides of an interaction, which is essential to induce the desirable learning dynamics observed in GCF.

**Property 2.** *NT-SSM induces type-dependent weight update dynamics across different neighbor pair types.*

We have shown that SSM applies a shared neighbor pair up-

*Table 2.* Each design choice in NT-BPR and NT-SSM contributes to its improved performance (in terms of NDCG@20).

|  | LastFM | MovieLens | Yelp | Amazon |
|---|---|---|---|---|
| BPR | 0.2530 | 0.2953 | 0.0449 | 0.0273 |
| w/o $\mathcal{L}(i;u)$ | 0.2459 | 0.2968 | 0.0421 | 0.0290 |
| w/o $\mathcal{L}(u;i)$ | 0.2593 | 0.3011 | 0.0477 | 0.0270 |
| w/o $\alpha$ | 0.2579 | 0.3008 | 0.0448 | 0.0287 |
| NT-BPR | **0.2654** | **0.3154** | **0.0480** | **0.0301** |
| SSM | 0.2404 | 0.2648 | 0.0530 | 0.0414 |
| w/o $\mathcal{L}(i;u)$ | 0.2696 | 0.2885 | 0.0551 | 0.0410 |
| w/o $\mathcal{L}(u;i)$ | 0.2494 | 0.2729 | 0.0559 | 0.0420 |
| w/o $\alpha$ | 0.2406 | 0.2677 | 0.0534 | 0.0411 |
| NT-SSM | **0.2709** | **0.3216** | **0.0565** | **0.0421** |

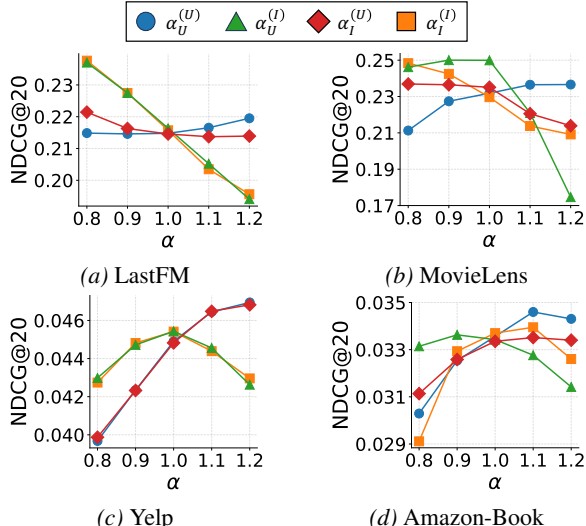

*Figure 5.* The controllable hyperparameters $\alpha_U^{(U)}$, $\alpha_U^{(I)}$, $\alpha_I^{(I)}$, and $\alpha_I^{(I)}$ of NT-SSM exhibit varying impacts across datasets, indicating the importance of flexible and adaptive adjustment.

weighting rule across all neighbor pair types (Limitation 2), despite our empirical findings that different types exhibit distinct optimal behaviors (Observation 2). NT-SSM addresses this issue by introducing neighbor-type–specific coefficients that modulate similarity updates differently across types. This enables the model to adaptively adjust how similarity updates are applied to different neighbor pair types, allowing the induced learning dynamics to better reflect their distinct roles in GCF.

### 6.4. Computational Cost of NT-SSM

NT-SSM has the same graph propagation cost as SSM, i.e., $O(L|\mathcal{E}|d)$ per epoch, and leaves inference cost unchanged because it only modifies the training objective. For loss computation, SSM costs $O(|\mathcal{E}||\mathcal{B}_u|d)$ per epoch, where $\mathcal{B}_u$ is the set of negative items sampled for user $u$. NT-SSM additionally includes the symmetric item-to-user contrastive term using negative users $\mathcal{B}_i$ and type-decomposed similarities, resulting in $O(|\mathcal{E}|(|\mathcal{B}_u|+|\mathcal{B}_i|)d)$ per epoch. When $|\mathcal{B}_u|$ and $|\mathcal{B}_i|$ are of the same order, NT-SSM preserves the same asymptotic order as SSM, with a larger constant factor.

## 7. Experimental Results

In this section, we evaluate the accuracy and effectiveness of NT-SSM by answering the following research questions:

- **Q1. Accuracy.** Does NT-SSM improve recommendation accuracy compared to SSM for GCF?
- **Q2. Effectiveness.** How do design components of NT-SSM contribute to performance gains?
- **Q3. Parameter Analysis.** How important is the adaptive adjustment of the neighbor-type coefficients in NT-SSM?
- **Q4. Efficiency.** What are the computational and memory overheads of NT-SSM compared with SSM?

### 7.1. Experimental Settings

We use four public datasets: LastFM (Iván et al., 2011), MovieLens-1M (Harper & Konstan, 2015), Yelp (He et al., 2020; Lin et al., 2022), and, Amazon-Book (He et al., 2020; Lin et al., 2022). Refer to Appendix E for the statistics of each dataset. Each dataset is split into training, validation, and test sets with a ratio of 7:1:2. We implemented NT-SSM based on the open-source library for GCF (Yu et al., 2023). We use cosine similarity for NT-SSM, following the standard SSM setting. The extension of NT-SSM to cosine similarity is detailed in Appendix D. The key hyperparameters of NT-SSM, $\alpha_I^{(U)}$, $\alpha_I^{(I)}$, $\alpha_U^{(U)}$, and $\alpha_U^{(I)}$ are tuned over $[0.5, 1.5]$ using Optuna (Akiba et al., 2019). We evaluate recommendation performance using NDCG@$N$ and Recall@$N$, averaged over five runs. We use $N = 20$, and results for $N = 10$ and $N = 40$ are in Appendix E.

### 7.2. Q1. Accuracy

First, we evaluate the recommendation performance of GCF models, specifically LightGCN (He et al., 2020), SimGCL (Yu et al., 2022), and NCL (Lin et al., 2022), trained with different optimization objectives. As shown in Table 1, NT-SSM consistently outperforms SSM, and NT-BPR consistently improves upon BPR across all datasets. For example, on MovieLens, NT-SSM achieves a 21.45% improvement over SSM, while NT-BPR yields a 6.81% improvement over BPR in terms of NDCG@20. These gains are observed across LightGCN, SimGCL, and NCL. We also observe that the improvements on SimGCL and NCL are relatively smaller than those on LightGCN, likely because these methods already introduce additional contrastive signals that partially reshape neighbor-pair interac-

*Table 3.* Efficiency comparison between SSM and NT-SSM. Time denotes average training time per epoch in seconds, and memory denotes peak GPU memory usage in MB.

| Dataset | Time (sec/epoch) | | Memory (MB) | |
|---|---|---|---|---|
| | SSM | NT-SSM | SSM | NT-SSM |
| LastFM | 0.47 | 0.73 | 42.9 | 47.7 |
| ML-1M | 6.16 | 8.70 | 54.7 | 56.9 |
| Yelp | 16.81 | 22.68 | 163.0 | 179.2 |
| Amazon-Book | 39.91 | 50.05 | 295.4 | 329.9 |

tions. Nevertheless, NT-SSM still provides consistent gains, suggesting that it is complementary to existing contrastive GCF frameworks. Overall, these results show that explicitly controlling weight update dynamics over neighbor pairs leads to improved GCF optimization.

### 7.3. Q2. Effectiveness

In Table 2, we conduct an ablation study to assess the effectiveness of each design choice in NT-BPR and NT-SSM. Specifically, we consider variants that (1) remove $\widetilde{\mathcal{L}}(i; u)$, (2) remove $\widetilde{\mathcal{L}}(u; i)$, and (3) fix all neighbor-type coefficients to $\alpha = 1$. Removing either $\widetilde{\mathcal{L}}(i; u)$ or $\widetilde{\mathcal{L}}(u; i)$ consistently degrades performance, indicating that the bidirectional formulation is crucial for effective recommendation. In addition, fixing the neighbor-type coefficients also leads to performance drops, confirming the importance of adaptively controlling neighbor pair coefficients according to their types. Overall, NT-SSM outperforms all ablated variants, validating the effectiveness of its design choices.

### 7.4. Q3. Parameter Analysis

We examine the effect of the controllable hyperparameters of NT-SSM on the performance of LightGCN. In Figure 5, we vary the four neighbor-type-specific coefficients, $\alpha_U^{(U)}$, $\alpha_U^{(I)}$, $\alpha_I^{(U)}$, and $\alpha_I^{(I)}$, within $\{0.8, \ldots, 1.2\}$ while fixing the other coefficients to their selected values. The results show that the effect of each coefficient varies across datasets and neighbor pair types. This supports our analysis that different neighbor pair types require different degrees of upweighting control, rather than a single uniform adjustment. In particular, the optimal settings are more type-dependent on denser or more skewed datasets, while larger and sparser datasets tend to prefer stronger filtering across multiple types, highlighting the importance of flexible, type-specific adjustment for improving recommendation performance. The selected optimal values for each dataset are reported in Appendix E.

### 7.5. Q4. Efficiency Analysis

Table 3 reports the average training time per epoch and peak GPU memory usage. [3] Consistent with the cost analysis in

---

[3] We used RTX 3090 Ti GPUs.

Section 6.4, NT-SSM incurs moderate training-time overhead and marginal memory overhead, while introducing no additional learnable parameters. The four neighbor-type-specific coefficients are tuned with Optuna using 100 trials per dataset, which is only 1% of a $10^4$ exhaustive grid over four coefficients with 10 candidate values each. The relative training-time overhead decreases on larger datasets; for example, on ML-1M, NT-SSM takes 8.70 seconds per epoch compared with 6.16 seconds for SSM, while improving NDCG@20 by 21.45%.

## 8. Conclusions, Limitations, and Future Work

In this work, we examine CL in GCF through the lens of its prediction mechanism. By unfolding the GCF prediction function, we show that user-item prediction scores are computed by aggregating learnable weights over a large number of multi-hop neighbor pairs, implying that effective optimization depends on which neighbor pairs are upweighted during training. Empirically, we find that effective recommendation is achieved by selectively upweighting neighbor pairs whose constituent neighbors are structurally similar to the target user and item, and that the impact of this selective upweighting varies across different neighbor pair types. Building on these findings, we revisit the learning dynamics of SSM and identify key limitations in its neighbor pair upweighting behavior. To address these limitations, we propose NT-SSM, a CL objective that exhibits desirable learning dynamics in GCF via type-aware neighbor pair upweighting. Experiments across multiple datasets and GCF models confirm the effectiveness of NT-SSM.

**Limitations.** Our theoretical analysis relies on LightGCN's linear aggregation and inner-product scoring. While the core insight—interpreting GCF learning as selective neighbor pair upweighting—conceptually extends beyond these assumptions, the exact mathematical unfolding may not readily generalize to complex scorers such as MLPs.

**Future work.** First, NT-SSM could be extended to operate during inference, as recent work has shown that training-free message passing and post-training embedding refinement can improve recommendation performance (Lee et al., 2024a; Ju et al., 2024). Second, recent studies have shown that data augmentation and training distribution design significantly affect recommendation performance (Liu et al., 2024; Lee et al., 2026), suggesting that reshaping the training distribution or graph connectivity could provide an orthogonal mechanism for controlling neighbor pair weighting. Third, extending NT-SSM beyond bipartite recommendation graphs to more complex relational settings, such as heterogeneous or multimodal graphs, may further improve its applicability and effectiveness (Tao et al., 2020).

## Acknowledgements

This work was partly supported by the National Research Foundation of Korea (NRF) grant funded by the Korea government (MSIT) (No. RS-2024-00406985, 30%). This work was partly supported by Institute of Information & Communications Technology Planning & Evaluation (IITP) grant funded by the Korea government (MSIT) (No. RS-2024-00438638, EntireDB2AI: Foundations and Software for Comprehensive Deep Representation Learning and Prediction on Entire Relational Databases, 30%) (No. RS-2022-II220157, Robust, Fair, Extensible Data-Centric Continual Learning, 30%) (No. RS-2019-II190075, Artificial Intelligence Graduate School Program (KAIST), 10%).

## Impact Statement

This paper aims to advance recommender systems by improving the optimization of graph collaborative filtering (GCF) through a neighbor type-aware contrastive learning (CL) objective. As with recommender systems in general, the deployment of such techniques should carefully consider potential societal impacts, including popularity bias and user privacy. In particular, care should be taken to ensure that unpopular users or items are not systematically overlooked, and that sensitive interaction data is not inappropriately propagated through the graph.

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

# A. Derivation of SSM Gradient w.r.t. Neighbor Pair Weight

In this section, we provide a complete derivation showing how contrastive learning with the SSM loss updates the learnable weights of individual neighbor pairs in GCF. We first review the SSM loss, compute the gradients with respect to similarity scores, recall the GCF similarity decomposition, and finally combine all terms to obtain a closed-form gradient expression.

**Setup: SSM Loss and Chain-Rule Decomposition.** We begin by reviewing the Sampled Softmax (SSM) loss.

For an observed interaction $(u, i) \in \mathcal{E}$, the SSM loss is defined as:

$$\mathcal{L}(i; u) = -\log \frac{\exp\left(s(u, i)/\tau\right)}{\exp\left(s(u, i)/\tau\right) + \sum_{j \in \mathcal{B}_u} \exp\left(s(u, j)/\tau\right)} = -\frac{s(u, i)}{\tau} + \log Z_u,$$

where $\mathcal{B}_u$ denotes the set of negative items sampled for user $u$, $\tau > 0$ is the temperature parameter, and

$$Z_u = \exp\left(s(u, i)/\tau\right) + \sum_{j \in \mathcal{B}_u} \exp\left(s(u, j)/\tau\right)$$

is the normalization term. We derive the gradient of $\mathcal{L}(i; u)$ with respect to the inner product $\mathbf{e}_v^{(0)\top} \mathbf{e}_{v'}^{(0)}$, which is the learnable weight of the neighbor pair $(v, v')$. Applying the chain rule over every item $x \in \{i\} \cup \mathcal{B}_u$ yields:

$$\frac{\partial \mathcal{L}(i; u)}{\partial(\mathbf{e}_v^{(0)\top} \mathbf{e}_{v'}^{(0)})} = \sum_{x \in \{i\} \cup \mathcal{B}_u} \frac{\partial \mathcal{L}(i; u)}{\partial s(u, x)} \cdot \frac{\partial s(u, x)}{\partial(\mathbf{e}_v^{(0)\top} \mathbf{e}_{v'}^{(0)})}. \tag{7}$$

**Step 1: Differentiating the Loss w.r.t. Similarity Scores.** First, we compute the gradients of the SSM loss with respect to the similarity scores of the positive and negative items.

For any negative item $j \in \mathcal{B}_u$, differentiating the loss yields:

$$\frac{\partial \mathcal{L}(i; u)}{\partial s(u, j)} = \frac{1}{\tau} \cdot \frac{\exp(s(u, j)/\tau)}{Z_u}.$$

For the positive item $i$, we obtain:

$$\frac{\partial \mathcal{L}(i; u)}{\partial s(u, i)} = \frac{1}{\tau} \left( \frac{\exp(s(u, i)/\tau)}{Z_u} - 1 \right).$$

We define the model-induced distribution over the positive and negative items as:

$$\pi_u(x) = \frac{\exp(s(u, x)/\tau)}{Z_u}, \qquad x \in \{i\} \cup \mathcal{B}_u.$$

Using this notation, the gradients can be written compactly as:

$$\frac{\partial \mathcal{L}(i; u)}{\partial s(u, j)} = \frac{1}{\tau} \pi_u(j), \qquad \frac{\partial \mathcal{L}(i; u)}{\partial s(u, i)} = \frac{1}{\tau} \big( \pi_u(i) - 1 \big). \tag{8}$$

**Step 2: Differentiating Similarity Scores w.r.t. Neighbor Pair Weights.** We then recall the decomposition of user–item similarity in GCF, where each similarity score is expressed as a weighted sum over neighbor pairs.

In GCF, the user–item similarity admits the following decomposition:

$$s(u, x) = \sum_{v \in \widetilde{\mathcal{N}}_u} \sum_{v' \in \widetilde{\mathcal{N}}_x} \widetilde{\mathbf{S}}_{uv} \cdot \widetilde{\mathbf{S}}_{xv'} \cdot \big( \mathbf{e}_v^{(0)\top} \mathbf{e}_{v'}^{(0)} \big), \qquad x \in \{i\} \cup \mathcal{B}_u,$$

where $\widetilde{\mathcal{N}}_u$ and $\widetilde{\mathcal{N}}_x$ denote the multi-hop neighborhoods of $u$ and $x$, respectively, and $\widetilde{\mathbf{S}}$ represents the structural weights induced by graph propagation. Since the similarity is linear in each neighbor-pair weight, fixing a particular pair $(v, v')$ and differentiating eliminates all other terms, yielding:

$$\frac{\partial s(u, x)}{\partial(\mathbf{e}_v^{(0)\top}\mathbf{e}_{v'}^{(0)})} = \widetilde{\mathbf{S}}_{uv}\widetilde{\mathbf{S}}_{xv'}. \tag{9}$$

**Step 3: Assembling the Full Gradient.** We now have all the ingredients to assemble the full gradient. Substituting the score-level gradients from Eq. (8) and the structural derivatives from Eq. (9) into the chain-rule expansion in Eq. (7), and separating the positive and negative contributions, we obtain

$$\frac{\partial \mathcal{L}(i; u)}{\partial(\mathbf{e}_v^{(0)\top}\mathbf{e}_{v'}^{(0)})} = \sum_{j \in \mathcal{B}_u} \frac{1}{\tau}\pi_u(j)\,\widetilde{\mathbf{S}}_{uv}\,\widetilde{\mathbf{S}}_{jv'} \;+\; \frac{1}{\tau}\big(\pi_u(i) - 1\big)\,\widetilde{\mathbf{S}}_{uv}\,\widetilde{\mathbf{S}}_{iv'} \qquad \text{(substitution)}$$

$$= \frac{\widetilde{\mathbf{S}}_{uv}}{\tau}\left(\sum_{j \in \mathcal{B}_u} \pi_u(j)\,\widetilde{\mathbf{S}}_{jv'} \;+\; \pi_u(i)\,\widetilde{\mathbf{S}}_{iv'} \;-\; \widetilde{\mathbf{S}}_{iv'}\right) \qquad \text{(factor out } \widetilde{\mathbf{S}}_{uv}/\tau)$$

$$= \frac{\widetilde{\mathbf{S}}_{uv}}{\tau}\left(\sum_{x \in \{i\} \cup \mathcal{B}_u} \pi_u(x)\,\widetilde{\mathbf{S}}_{xv'} \;-\; \widetilde{\mathbf{S}}_{iv'}\right). \qquad \text{(merge sums)}$$

The summation term admits a natural probabilistic interpretation: it is the expectation of the structural weight $\widetilde{\mathbf{S}}_{xv'}$ under the model's current distribution $\pi_u$ over the candidate set $\{i\} \cup \mathcal{B}_u$:

$$\mathbb{E}_{x \sim \pi_u}\big[\widetilde{\mathbf{S}}_{xv'}\big] \;=\; \sum_{x \in \{i\} \cup \mathcal{B}_u} \pi_u(x)\,\widetilde{\mathbf{S}}_{xv'}.$$

Substituting this expectation into the gradient expression, we arrive at the final closed-form result:

$$\boxed{\frac{\partial \mathcal{L}(i; u)}{\partial(\mathbf{e}_v^{(0)\top}\mathbf{e}_{v'}^{(0)})} \;=\; \frac{\widetilde{\mathbf{S}}_{uv}}{\tau}\left(\mathbb{E}_{x \sim \pi_u}\big[\widetilde{\mathbf{S}}_{xv'}\big] \;-\; \widetilde{\mathbf{S}}_{iv'}\right)}.$$

## B. Derivation of NT-SSM Gradient w.r.t. Neighbor Pair Weight

In this section, we provide a complete derivation showing how the proposed NT-SSM loss updates the learnable weights of individual neighbor pairs. We first define the bidirectional objective and focus on the user-to-item direction, then apply the chain rule, compute each factor, and assemble the full gradient before combining both directions into a unified update rule.

**Setup: NT-SSM Loss and Chain-Rule Decomposition.** NT-SSM optimizes a bidirectional contrastive loss that aggregates signal from both user-to-item and item-to-user directions. For an observed interaction $(u, i) \in \mathcal{E}$, the loss is:

$$\mathcal{L}(u, i) = \widetilde{\mathcal{L}}(i; u) + \widetilde{\mathcal{L}}(u; i),$$

where the two terms contrast items conditioned on the user and users conditioned on the item, respectively.

Without loss of generality, we focus on the user-to-item component $\widetilde{\mathcal{L}}(i; u)$; the derivation for $\widetilde{\mathcal{L}}(u; i)$ follows by symmetry and is incorporated in the final step. The user-to-item loss is given by

$$\widetilde{\mathcal{L}}(i; u) = -\log \frac{\exp(s(u, i)/\tau)}{\exp(s(u, i)/\tau) + \sum_{j \in \mathcal{B}_u} \exp(\widetilde{s}(u, j)/\tau)} = -\frac{s(u, i)}{\tau} + \log \widetilde{Z}_u,$$

where $\mathcal{B}_u$ denotes the set of negative items sampled for user $u$, and

$$\widetilde{Z}_u = \exp(s(u, i)/\tau) + \sum_{j \in \mathcal{B}_u} \exp(\widetilde{s}(u, j)/\tau)$$

is the partition term. A key distinction from standard SSM is that, for negative items $j \in \mathcal{B}_u$, the similarity score $\widetilde{s}(u, j)$ is re-weighted by neighbor-type-specific coefficients:

$$\widetilde{s}(u, j) = \alpha_I^{(U)} s_I^{(U)}(u, j) + \alpha_I^{(I)} s_I^{(I)}(u, j).$$

Our goal is to derive the gradient of the loss with respect to the inner product $\mathbf{e}_v^{(0)\top} \mathbf{e}_{v'}^{(0)}$ of a specific neighbor pair $(v, v')$, where $v'$ belongs to type $t_{v'} \in \{U, I\}$. Its contribution to the negative similarity is scaled by the coefficient $\alpha_I^{(t_{v'})}$.

Applying the chain rule, we decompose this gradient into contributions from the positive item and each negative item:

$$\frac{\partial \widetilde{\mathcal{L}}(i; u)}{\partial(\mathbf{e}_v^{(0)\top} \mathbf{e}_{v'}^{(0)})} = \sum_{j \in \mathcal{B}_u} \frac{\partial \widetilde{\mathcal{L}}(i; u)}{\partial \widetilde{s}(u, j)} \frac{\partial \widetilde{s}(u, j)}{\partial(\mathbf{e}_v^{(0)\top} \mathbf{e}_{v'}^{(0)})} + \frac{\partial \widetilde{\mathcal{L}}(i; u)}{\partial s(u, i)} \frac{\partial s(u, i)}{\partial(\mathbf{e}_v^{(0)\top} \mathbf{e}_{v'}^{(0)})}. \tag{10}$$

Note that, unlike standard SSM, the negative terms differentiate through the type-weighted score $\widetilde{s}(u, j)$ rather than the raw score $s(u, j)$. We compute each factor in the following two steps.

**Step 1: Differentiating the Loss w.r.t. Similarity Scores.** We define the model-induced distribution over the candidate set, accounting for the asymmetry between positive and negative scores:

$$\pi_u(x) = \begin{cases} \dfrac{\exp(s(u, x)/\tau)}{\widetilde{Z}_u} & \text{if } x = i, \\ \dfrac{\exp(\widetilde{s}(u, x)/\tau)}{\widetilde{Z}_u} & \text{if } x \in \mathcal{B}_u. \end{cases}$$

Note that, unlike standard SSM, the positive and negative cases use different similarity functions ($s$ vs. $\widetilde{s}$), reflecting the type-specific reweighting introduced by NT-SSM.

Since the loss decomposes as $\widetilde{\mathcal{L}}(i; u) = -s(u, i)/\tau + \log \widetilde{Z}_u$, we differentiate each term separately. For any negative item $j \in \mathcal{B}_u$, the first term $-s(u, i)/\tau$ is independent of $\widetilde{s}(u, j)$, so only the $\log \widetilde{Z}_u$ term contributes:

$$\frac{\partial \widetilde{\mathcal{L}}(i; u)}{\partial \widetilde{s}(u, j)} = \frac{1}{\widetilde{Z}_u} \cdot \frac{1}{\tau} \exp(\widetilde{s}(u, j)/\tau) = \frac{1}{\tau} \pi_u(j).$$

For the positive item $i$, both terms contribute: the $\log \widetilde{Z}_u$ term yields the same softmax form, while the $-s(u, i)/\tau$ term adds an extra $-1/\tau$, giving

$$\frac{\partial \widetilde{\mathcal{L}}(i; u)}{\partial s(u, i)} = -\frac{1}{\tau} + \frac{1}{\widetilde{Z}_u} \cdot \frac{1}{\tau} \exp(s(u, i)/\tau) = \frac{1}{\tau} \big(\pi_u(i) - 1\big).$$

Using this notation, the gradients can be written compactly as:

$$\frac{\partial \widetilde{\mathcal{L}}(i; u)}{\partial \widetilde{s}(u, j)} = \frac{1}{\tau} \pi_u(j), \qquad \frac{\partial \widetilde{\mathcal{L}}(i; u)}{\partial s(u, i)} = \frac{1}{\tau} \big(\pi_u(i) - 1\big). \tag{11}$$

**Step 2: Differentiating Similarity Scores w.r.t. Neighbor Pair Weights.** The structural derivatives differ for positive and negative items due to the type-specific reweighting in NT-SSM.

For a negative item $j$, however, the type-weighted similarity $\widetilde{s}(u, j)$ causes the derivative to pick up the coefficient $\alpha_I^{(t_{v'})}$ corresponding to the type of neighbor $v'$:

$$\frac{\partial \widetilde{s}(u, j)}{\partial(\mathbf{e}_v^{(0)\top} \mathbf{e}_{v'}^{(0)})} = \alpha_I^{(t_{v'})} \cdot \widetilde{\mathbf{S}}_{uv} \widetilde{\mathbf{S}}_{jv'}.$$

This asymmetry is the mechanism through which NT-SSM introduces type-aware control into the gradient dynamics.

For the positive item $i$, the standard GCF decomposition applies (cf. Appendix A):

$$\frac{\partial s(u,i)}{\partial(\mathbf{e}_v^{(0)\top}\mathbf{e}_{v'}^{(0)})} = \widetilde{\mathbf{S}}_{uv}\,\widetilde{\mathbf{S}}_{iv'}.$$

Using this notation, the gradients can be written compactly as:

$$\frac{\partial\widetilde{s}(u,j)}{\partial(\mathbf{e}_v^{(0)\top}\mathbf{e}_{v'}^{(0)})} = \alpha_I^{(t_{v'})}\cdot\widetilde{\mathbf{S}}_{uv}\,\widetilde{\mathbf{S}}_{jv'}, \qquad \frac{\partial s(u,i)}{\partial(\mathbf{e}_v^{(0)\top}\mathbf{e}_{v'}^{(0)})} = \widetilde{\mathbf{S}}_{uv}\,\widetilde{\mathbf{S}}_{iv'}. \tag{12}$$

**Step 3: Assembling the User-to-Item Gradient.** By substituting Eq. (11) and Eq. (12) into the chain-rule expansion (Eq. (10)), we obtain:

$$\frac{\partial\widetilde{\mathcal{L}}(i;u)}{\partial(\mathbf{e}_v^{(0)\top}\mathbf{e}_{v'}^{(0)})} = \sum_{j\in\mathcal{B}_u}\frac{1}{\tau}\,\pi_u(j)\cdot\alpha_I^{(t_{v'})}\,\widetilde{\mathbf{S}}_{uv}\,\widetilde{\mathbf{S}}_{jv'} \;+\; \frac{1}{\tau}\big(\pi_u(i)-1\big)\,\widetilde{\mathbf{S}}_{uv}\,\widetilde{\mathbf{S}}_{iv'} \qquad \text{(substitution)}$$

$$= \frac{\widetilde{\mathbf{S}}_{uv}}{\tau}\left(\alpha_I^{(t_{v'})}\sum_{j\in\mathcal{B}_u}\pi_u(j)\,\widetilde{\mathbf{S}}_{jv'} \;-\; (1-\pi_u(i))\,\widetilde{\mathbf{S}}_{iv'}\right). \qquad \text{(factor out } \widetilde{\mathbf{S}}_{uv}/\tau)$$

Let $\Pi_u = \sum_{j\in\mathcal{B}_u}\pi_u(j) = 1-\pi_u(i)$ denote the total probability mass assigned to the negative items, and define the conditional distribution $\hat{\pi}_u(j) = \pi_u(j)/\Pi_u$ over the negative set $\mathcal{B}_u$. Rewriting the sum as $\Pi_u\,\mathbb{E}_{j\sim\hat{\pi}_u}[\widetilde{\mathbf{S}}_{jv'}]$ and factoring out $\Pi_u$ from both terms, we arrive at:

$$\frac{\partial\widetilde{\mathcal{L}}(i;u)}{\partial(\mathbf{e}_v^{(0)\top}\mathbf{e}_{v'}^{(0)})} = \frac{\Pi_u\,\widetilde{\mathbf{S}}_{uv}}{\tau}\left(\alpha_I^{(t_{v'})}\cdot\mathbb{E}_{j\sim\hat{\pi}_u}[\widetilde{\mathbf{S}}_{jv'}] \;-\; \widetilde{\mathbf{S}}_{iv'}\right).$$

**Step 4: Bidirectional Aggregation.** The total gradient combines the user-to-item and item-to-user components by linearity of differentiation:

$$\frac{\partial\mathcal{L}(u,i)}{\partial(\mathbf{e}_v^{(0)\top}\mathbf{e}_{v'}^{(0)})} = \frac{\partial\widetilde{\mathcal{L}}(i;u)}{\partial(\mathbf{e}_v^{(0)\top}\mathbf{e}_{v'}^{(0)})} + \frac{\partial\widetilde{\mathcal{L}}(u;i)}{\partial(\mathbf{e}_v^{(0)\top}\mathbf{e}_{v'}^{(0)})}.$$

The first term governs updates through the user-to-item contrastive task, where the gradient is modulated by the *item-side* neighbor type of $v'$ through the coefficient $\alpha_I^{(t_{v'})}$. Symmetrically, the second term governs updates through the item-to-user contrastive task, where the gradient is modulated by the *user-side* neighbor type of $v$ through the coefficient $\alpha_U^{(t_v)}$.

Let $\Pi_i = 1-\pi_i(u)$ denote the total probability mass assigned to negative users, and let $\hat{\pi}_i$ be the corresponding conditional distribution over $\mathcal{B}_i$. Combining the boxed result from Step 4 with its symmetric counterpart, we obtain the final unified update rule:

$$\boxed{\frac{\partial\mathcal{L}(u,i)}{\partial(\mathbf{e}_v^{(0)\top}\mathbf{e}_{v'}^{(0)})} = \underbrace{\frac{\Pi_u\,\widetilde{\mathbf{S}}_{uv}}{\tau}\left(\alpha_I^{(t_{v'})}\,\mathbb{E}_{j\sim\hat{\pi}_u}[\widetilde{\mathbf{S}}_{jv'}]-\widetilde{\mathbf{S}}_{iv'}\right)}_{\text{user-to-item direction}} + \underbrace{\frac{\Pi_i\,\widetilde{\mathbf{S}}_{iv'}}{\tau}\left(\alpha_U^{(t_v)}\,\mathbb{E}_{k\sim\hat{\pi}_i}[\widetilde{\mathbf{S}}_{kv}]-\widetilde{\mathbf{S}}_{uv}\right)}_{\text{item-to-user direction}}.}$$

## C. NT-BPR: Neighbor Type-Aware BPR Loss

The Bayesian Personalized Ranking (BPR) loss (**?**) is a widely used contrastive objective in GCF, which adopts a pairwise formulation. Given a user $u$, a positive item $i$, and a negative item $j$, the BPR loss is defined as:

$$\mathcal{L}_{\text{BPR}}(i,j;u) = -\log\sigma(s(u,i)-s(u,j)),$$

where $\sigma(\cdot)$ is the sigmoid function.

Our framework can be readily extended to BPR by introducing neighbor-type–aware similarity updates for negative items, analogous to our enhancement of SSM. Specifically, we define the neighbor-type–aware BPR (NT-BPR) loss as:

$$\widetilde{\mathcal{L}}_{\text{BPR}}(i;u) = -\log \sigma(s(u,i) - \widetilde{s}(u,j)),$$

where the negative similarity $\widetilde{s}(u,j)$ is decomposed by neighbor types using type-specific coefficients, following the same principle as in NT-SSM.

## D. Extension to Cosine Similarity

In Appendices A and B, we derived the gradients of SSM and NT-SSM under the inner product similarity $s(u,i) = \mathbf{e}_u^\top \mathbf{e}_i$. Here, we show that the analysis extends to cosine similarity $s_{\cos}(u,i) = \mathbf{e}_u^\top \mathbf{e}_i / (\|\mathbf{e}_u\|\|\mathbf{e}_i\|)$.

**Cosine Similarity Decomposition.** Recall from Section 3 that the final embedding decomposes as $\mathbf{e}_x = \mathbf{e}_x^{(\mathcal{U})} + \mathbf{e}_x^{(\mathcal{I})}$, where $\mathbf{e}_x^{(t)} = \sum_{v \in \widetilde{\mathcal{N}}_x^{(t)}} \widetilde{\mathbf{S}}_{xv} \mathbf{e}_v^{(0)}$ for $t \in \{\mathcal{U}, \mathcal{I}\}$. Expanding the numerator of $s_{\cos}(u,i)$ yields four bilinear terms:

$$s_{\cos}(u,i) = \sum_{t,t' \in \{\mathcal{U},\mathcal{I}\}} C^{(t,t')}(u,i), \quad \text{where} \quad C^{(t,t')}(u,i) = \frac{\mathbf{e}_u^{(t)\top} \mathbf{e}_i^{(t')}}{\|\mathbf{e}_u\| \cdot \|\mathbf{e}_i\|}.$$

Compared to the inner product decomposition (Eq. (2)), each neighbor pair weight $\mathbf{e}_v^{(0)\top} \mathbf{e}_{v'}^{(0)}$ is additionally scaled by the shared normalization factor $(\|\mathbf{e}_u\|\|\mathbf{e}_i\|)^{-1}$, which does not depend on the specific pair $(v, v')$.

**NT-SSM under Cosine Similarity.** For a negative item $j \in \mathcal{B}_u$, the type-aware negative similarity becomes:

$$\widetilde{s}_{\cos}(u,j) = \sum_{t,t' \in \{\mathcal{U},\mathcal{I}\}} \alpha_{\mathcal{I}}^{(t')} C^{(t,t')}(u,j),$$

where the coefficient $\alpha_{\mathcal{I}}^{(t')}$ is determined by the item-side neighbor type $t'$, as in Section 6.1.

**Gradient Derivation.** Following the same chain-rule decomposition as in Appendix B, we compute the gradient of $\widetilde{\mathcal{L}}(i;u)$ with respect to the weight of a neighbor pair $(v, v')$ with $v \in \widetilde{\mathcal{N}}_u$ and $v' \in \widetilde{\mathcal{N}}_i$. The loss-to-score gradients remain identical to Eq. (11). The score-to-weight gradients differ only by the normalization factor: for the positive item, $\partial s_{\cos}(u,i)/\partial(\mathbf{e}_v^{(0)\top} \mathbf{e}_{v'}^{(0)}) = \widetilde{\mathbf{S}}_{uv} \widetilde{\mathbf{S}}_{iv'}/(\|\mathbf{e}_u\|\|\mathbf{e}_i\|)$, and for a negative item $j$, $\partial \widetilde{s}_{\cos}(u,j)/\partial(\mathbf{e}_v^{(0)\top} \mathbf{e}_{v'}^{(0)}) = \alpha_{\mathcal{I}}^{(t_{v'})} \widetilde{\mathbf{S}}_{uv} \widetilde{\mathbf{S}}_{jv'}/(\|\mathbf{e}_u\|\|\mathbf{e}_j\|)$, where we treat the norms as approximately constant with respect to a single neighbor pair weight.[4] Assembling as in Appendix B, we obtain:

$$\frac{\partial \widetilde{\mathcal{L}}(i;u)}{\partial(\mathbf{e}_v^{(0)\top} \mathbf{e}_{v'}^{(0)})} = \frac{\Pi_u \widetilde{\mathbf{S}}_{uv}}{\tau} \left( \alpha_{\mathcal{I}}^{(t_{v'})} \mathbb{E}_{j \sim \hat{\pi}_u} \left[ \frac{\widetilde{\mathbf{S}}_{jv'}}{\|\mathbf{e}_u\|\|\mathbf{e}_j\|} \right] - \frac{\widetilde{\mathbf{S}}_{iv'}}{\|\mathbf{e}_u\|\|\mathbf{e}_i\|} \right).$$

Since $\|\mathbf{e}_u\|^{-1}$ is shared across both terms, it cancels, yielding the upweighting condition:

$$\frac{\widetilde{\mathbf{S}}_{iv'}}{\|\mathbf{e}_i\|} > \alpha_{\mathcal{I}}^{(t_{v'})} \mathbb{E}_{j \sim \hat{\pi}_u} \left[ \frac{\widetilde{\mathbf{S}}_{jv'}}{\|\mathbf{e}_j\|} \right].$$

This is the norm-adjusted counterpart of the inner product condition in Appendix B, where $\widetilde{\mathbf{S}}_{iv'}$ is replaced by $\widetilde{\mathbf{S}}_{iv'}/\|\mathbf{e}_i\|$. The bidirectional aggregation with $\widetilde{\mathcal{L}}(u;i)$ follows symmetrically, incorporating both user- and item-side structural similarities as in Step 4 of Appendix B.

## E. Additional Experimental Settings & Results

In this section, we provide additional experimental settings, results, and analyses omitted from the main paper due to space limitations.

---

[4]The partial derivative of $\|\mathbf{e}_u\|$ with respect to a single weight $\mathbf{e}_v^{(0)\top} \mathbf{e}_{v'}^{(0)}$ is $O(1/|\widetilde{\mathcal{N}}_u|)$, which is negligible given the large number of neighbor pairs (Figure 1).

*Table 4.* Dataset Statistics.

| Dataset | # User | # Item | # Interaction | Density |
|---|---|---|---|---|
| LastFM | 1,885 | 17,388 | 91,779 | 0.002800 |
| MovieLens | 6,039 | 3,628 | 836,478 | 0.038178 |
| Yelp | 31,668 | 38,048 | 1,561,406 | 0.001295 |
| Amazon-Book | 52,643 | 91,599 | 2,984,108 | 0.000619 |

*Table 5.* Optimal hyperparameter values of NT-BPR and NT-SSM on each dataset and backbone model.

| Model | Dataset | NT-BPR | | | | NT-SSM | | | |
|---|---|---|---|---|---|---|---|---|---|
| | | $\alpha_U^{(U)}$ | $\alpha_I^{(I)}$ | $\alpha_U^{(I)}$ | $\alpha_I^{(U)}$ | $\alpha_U^{(U)}$ | $\alpha_I^{(I)}$ | $\alpha_U^{(I)}$ | $\alpha_I^{(U)}$ |
| LightGCN | LastFM | 1.3 | 1.5 | 0.9 | 1.3 | 1.2 | 0.8 | 0.8 | 0.9 |
| | ML-1M | 1.4 | 1.3 | 0.8 | 1.3 | 1.2 | 0.8 | 0.8 | 1.0 |
| | Yelp | 1.5 | 1.5 | 1.4 | 1.5 | 1.2 | 1.2 | 1.1 | 1.2 |
| | Amazon | 1.5 | 1.5 | 1.4 | 1.5 | 1.2 | 1.1 | 1.1 | 1.1 |
| SimGCL | LastFM | 0.7 | 1.1 | 1.1 | 0.9 | 0.8 | 0.9 | 1.1 | 0.8 |
| | ML-1M | 0.8 | 0.7 | 0.8 | 1.5 | 0.8 | 0.9 | 0.9 | 0.8 |
| | Yelp | 1.2 | 1.0 | 0.8 | 1.0 | 1.2 | 1.3 | 1.4 | 0.7 |
| | Amazon | 1.2 | 0.8 | 0.8 | 0.8 | 1.3 | 1.1 | 0.9 | 1.3 |
| NCL | LastFM | 1.1 | 1.3 | 0.9 | 1.4 | 1.3 | 0.7 | 0.7 | 1.1 |
| | ML-1M | 1.2 | 1.2 | 0.8 | 1.2 | 1.2 | 0.8 | 0.8 | 1.2 |
| | Yelp | 1.3 | 1.3 | 1.3 | 0.9 | 0.8 | 1.0 | 0.8 | 1.0 |
| | Amazon | 1.4 | 0.6 | 1.4 | 1.4 | 1.0 | 1.0 | 1.0 | 1.0 |

**Dataset Statistics.** In Table 4, we provide the statistics of the datasets used in our experiments, including the numbers of users, items, interactions, and sparsity levels.

**Results at Different Cutoffs.** In Tables 6 and 7, we report the recommendation performance of GCF models trained with BPR, SSM, NT-BPR, and NT-SSM, evaluated using Recall/NDCG@10 and Recall/NDCG@40, respectively. Overall, NT-BPR and NT-SSM consistently outperform BPR and SSM, respectively, across most datasets and backbone models, often by substantial margins. These results demonstrate that the effectiveness of the proposed neighbor type-aware learning dynamics generalizes across different evaluation cutoffs.

**Optimal Hyperparameters.** In Table 5, we report the optimal hyperparameter configurations of NT-BPR and NT-SSM obtained on each dataset. The optimal coefficients vary substantially across datasets and neighbor pair types, further supporting our observation that different neighbor pair types exhibit distinct learning dynamics and should therefore be controlled adaptively.

*Table 6.* Overall recommendation performance of CL objectives, BPR and SSM, and their neighbor type-aware counterparts, NT-BPR and NT-SSM, respectively, under GCF models in terms of Recall@10 and NDCG@10.

| Method | Loss | LastFM | | MovieLens | | Yelp | | Amazon-Book | |
|---|---|---|---|---|---|---|---|---|---|
| | | Recall@10 | NDCG@10 | Recall@10 | NDCG@10 | Recall@10 | NDCG@10 | Recall@10 | NDCG@10 |
| LightGCN | BPR | $0.1898_{\pm0.0030}$ | $0.2080_{\pm0.0020}$ | $0.1458_{\pm0.0010}$ | $0.2928_{\pm0.0007}$ | $0.0321_{\pm0.0004}$ | $0.0362_{\pm0.0004}$ | $0.0204_{\pm0.0003}$ | $0.0213_{\pm0.0002}$ |
| | NT-BPR | $\mathbf{0.1996}_{\pm0.0031}$ | $\mathbf{0.2182}_{\pm0.0020}$ | $\mathbf{0.1579}_{\pm0.0010}$ | $\mathbf{0.3127}_{\pm0.0009}$ | $\mathbf{0.0344}_{\pm0.0004}$ | $\mathbf{0.0391}_{\pm0.0004}$ | $\mathbf{0.0227}_{\pm0.0001}$ | $\mathbf{0.0234}_{\pm0.0002}$ |
| | Improv. | 5.16% | 4.90% | 8.30% | 6.80% | 7.17% | 8.01% | 11.27% | 9.86% |
| | SSM | $0.1802_{\pm0.0006}$ | $0.1977_{\pm0.0010}$ | $0.1346_{\pm0.0011}$ | $0.2629_{\pm0.0033}$ | $0.0383_{\pm0.0002}$ | $0.0432_{\pm0.0003}$ | $0.0316_{\pm0.0002}$ | $0.0328_{\pm0.0003}$ |
| | NT-SSM | $\mathbf{0.2048}_{\pm0.0009}$ | $\mathbf{0.2235}_{\pm0.0013}$ | $\mathbf{0.1630}_{\pm0.0003}$ | $\mathbf{0.3203}_{\pm0.0003}$ | $\mathbf{0.0409}_{\pm0.0001}$ | $\mathbf{0.0461}_{\pm0.0001}$ | $\mathbf{0.0323}_{\pm0.0003}$ | $\mathbf{0.0334}_{\pm0.00023}$ |
| | Improv. | 13.65% | 13.05% | 21.10% | 21.83% | 6.79% | 6.72% | 2.22% | 1.83% |
| SimGCL | BPR | $0.2017_{\pm0.0015}$ | $0.2208_{\pm0.0010}$ | $0.1609_{\pm0.0011}$ | $0.3138_{\pm0.0007}$ | $0.0389_{\pm0.0004}$ | $0.0438_{\pm0.0003}$ | $0.0259_{\pm0.0002}$ | $0.0266_{\pm0.0002}$ |
| | NT-BPR | $\mathbf{0.2038}_{\pm0.0010}$ | $\mathbf{0.2221}_{\pm0.0008}$ | $\mathbf{0.1634}_{\pm0.0007}$ | $\mathbf{0.3162}_{\pm0.0016}$ | $\mathbf{0.0393}_{\pm0.0002}$ | $\mathbf{0.0442}_{\pm0.0002}$ | $\mathbf{0.0267}_{\pm0.0002}$ | $\mathbf{0.0274}_{\pm0.0001}$ |
| | Improv. | 1.04% | 0.59% | 1.55% | 0.76% | 1.03% | 0.91% | 3.09% | 3.01% |
| | SSM | $0.1929_{\pm0.0007}$ | $0.2114_{\pm0.0007}$ | $0.1495_{\pm0.0006}$ | $0.2862_{\pm0.0029}$ | $0.0400_{\pm0.0003}$ | $0.0454_{\pm0.0003}$ | $0.0268_{\pm0.0001}$ | $0.0280_{\pm0.0001}$ |
| | NT-SSM | $\mathbf{0.2044}_{\pm0.0008}$ | $\mathbf{0.2259}_{\pm0.0011}$ | $\mathbf{0.1671}_{\pm0.0011}$ | $\mathbf{0.3239}_{\pm0.0008}$ | $\mathbf{0.0403}_{\pm0.0001}$ | $\mathbf{0.0456}_{\pm0.0002}$ | $\mathbf{0.0287}_{\pm0.0002}$ | $\mathbf{0.0305}_{\pm0.0003}$ |
| | Improv. | 5.96% | 6.86% | 11.77% | 13.17% | 0.75% | 0.44% | 7.09% | 8.93% |
| NCL | BPR | $0.1988_{\pm0.0021}$ | $0.2186_{\pm0.0012}$ | $0.1555_{\pm0.0009}$ | $0.3094_{\pm0.0014}$ | $0.0362_{\pm0.0005}$ | $0.0407_{\pm0.0004}$ | $0.0228_{\pm0.0002}$ | $0.0234_{\pm0.0001}$ |
| | NT-BPR | $\mathbf{0.2017}_{\pm0.0010}$ | $\mathbf{0.2225}_{\pm0.0007}$ | $\mathbf{0.1590}_{\pm0.0007}$ | $\mathbf{0.3149}_{\pm0.0008}$ | $\mathbf{0.0367}_{\pm0.0002}$ | $\mathbf{0.0416}_{\pm0.0002}$ | $\mathbf{0.0253}_{\pm0.0002}$ | $\mathbf{0.0260}_{\pm0.0002}$ |
| | Improv. | 1.46% | 1.78% | 2.25% | 1.78% | 1.38% | 2.21% | 10.96% | 11.11% |
| | SSM | $0.1950_{\pm0.0010}$ | $0.2132_{\pm0.0010}$ | $0.1381_{\pm0.0012}$ | $0.2613_{\pm0.0021}$ | $0.0387_{\pm0.0002}$ | $0.0438_{\pm0.0001}$ | $0.0237_{\pm0.0001}$ | $0.0254_{\pm0.0001}$ |
| | NT-SSM | $\mathbf{0.2050}_{\pm0.0022}$ | $\mathbf{0.2231}_{\pm0.0018}$ | $\mathbf{0.1600}_{\pm0.0009}$ | $\mathbf{0.3111}_{\pm0.0007}$ | $\mathbf{0.0390}_{\pm0.0001}$ | $\mathbf{0.0442}_{\pm0.0003}$ | $\mathbf{0.0315}_{\pm0.0002}$ | $\mathbf{0.0326}_{\pm0.0002}$ |
| | Improv. | 5.13% | 4.64% | 15.86% | 19.06% | 0.78% | 0.91% | 32.91% | 28.35% |

*Table 7.* Overall recommendation performance of CL objectives, BPR and SSM, and their neighbor type-aware counterparts, NT-BPR and NT-SSM, respectively, under GCF models in terms of Recall@40 and NDCG@40.

| Method | Loss | LastFM | | MovieLens | | Yelp | | Amazon-Book | |
|---|---|---|---|---|---|---|---|---|---|
| | | Recall@40 | NDCG@40 | Recall@40 | NDCG@40 | Recall@40 | NDCG@40 | Recall@40 | NDCG@40 |
| LightGCN | BPR | $0.3835_{\pm0.0013}$ | $0.2986_{\pm0.0012}$ | $0.3533_{\pm0.0014}$ | $0.3216_{\pm0.0009}$ | $0.0927_{\pm0.0004}$ | $0.0591_{\pm0.0003}$ | $0.0599_{\pm0.0003}$ | $0.0366_{\pm0.0003}$ |
| | NT-BPR | $\mathbf{0.3982}_{\pm0.0034}$ | $\mathbf{0.3113}_{\pm0.0018}$ | $\mathbf{0.3700}_{\pm0.0011}$ | $\mathbf{0.3409}_{\pm0.0006}$ | $\mathbf{0.0978}_{\pm0.0007}$ | $\mathbf{0.0628}_{\pm0.0004}$ | $\mathbf{0.0647}_{\pm0.0006}$ | $\mathbf{0.0397}_{\pm0.0004}$ |
| | Improv. | 3.83% | 4.25% | 4.73% | 6.00% | 5.50% | 6.26% | 8.01% | 8.47% |
| | SSM | $0.3584_{\pm0.0014}$ | $0.2814_{\pm0.0009}$ | $0.3204_{\pm0.0015}$ | $0.2884_{\pm0.0024}$ | $0.1061_{\pm0.0003}$ | $0.0687_{\pm0.0001}$ | $0.0860_{\pm0.0003}$ | $0.0538_{\pm0.0003}$ |
| | NT-SSM | $\mathbf{0.4036}_{\pm0.0022}$ | $\mathbf{0.3169}_{\pm0.0017}$ | $\mathbf{0.3745}_{\pm0.0001}$ | $\mathbf{0.3462}_{\pm0.0002}$ | $\mathbf{0.1136}_{\pm0.0003}$ | $\mathbf{0.0733}_{\pm0.0001}$ | $\mathbf{0.0866}_{\pm0.0003}$ | $\mathbf{0.0544}_{\pm0.0002}$ |
| | Improv. | 12.61% | 12.62% | 16.89% | 20.04% | 7.07% | 6.70% | 0.70% | 1.12% |
| SimGCL | BPR | $0.4023_{\pm0.0014}$ | $0.3149_{\pm0.0007}$ | $0.3737_{\pm0.0018}$ | $0.3419_{\pm0.0009}$ | $0.1107_{\pm0.0005}$ | $0.0709_{\pm0.0002}$ | $0.0733_{\pm0.0007}$ | $0.0451_{\pm0.0004}$ |
| | NT-BPR | $\mathbf{0.4025}_{\pm0.0017}$ | $\mathbf{0.3154}_{\pm0.0012}$ | $\mathbf{0.3771}_{\pm0.0005}$ | $\mathbf{0.3446}_{\pm0.0009}$ | $\mathbf{0.1114}_{\pm0.0004}$ | $\mathbf{0.0715}_{\pm0.0003}$ | $\mathbf{0.0745}_{\pm0.0003}$ | $\mathbf{0.0461}_{\pm0.0001}$ |
| | Improv. | 0.05% | 0.16% | 0.91% | 0.79% | 0.63% | 0.85% | 1.64% | 2.22% |
| | SSM | $0.3851_{\pm0.0009}$ | $0.3013_{\pm0.0006}$ | $0.3510_{\pm0.0009}$ | $0.3161_{\pm0.0015}$ | $0.1110_{\pm0.0002}$ | $0.0719_{\pm0.0002}$ | $0.0755_{\pm0.0003}$ | $0.0468_{\pm0.0001}$ |
| | NT-SSM | $\mathbf{0.4044}_{\pm0.0021}$ | $\mathbf{0.3195}_{\pm0.0014}$ | $\mathbf{0.3815}_{\pm0.0011}$ | $\mathbf{0.3512}_{\pm0.0004}$ | $\mathbf{0.1117}_{\pm0.0003}$ | $\mathbf{0.0724}_{\pm0.0001}$ | $\mathbf{0.0796}_{\pm0.0003}$ | $\mathbf{0.0500}_{\pm0.0003}$ |
| | Improv. | 5.01% | 6.04% | 8.69% | 11.10% | 0.63% | 0.70% | 5.43% | 6.84% |
| NCL | BPR | $0.4013_{\pm0.0025}$ | $0.3136_{\pm0.0015}$ | $0.3658_{\pm0.0028}$ | $0.3364_{\pm0.0017}$ | $0.1022_{\pm0.0005}$ | $0.0656_{\pm0.0003}$ | $0.0651_{\pm0.0005}$ | $0.0398_{\pm0.0003}$ |
| | NT-BPR | $\mathbf{0.4025}_{\pm0.0023}$ | $\mathbf{0.3165}_{\pm0.0011}$ | $\mathbf{0.3707}_{\pm0.0013}$ | $\mathbf{0.3423}_{\pm0.0015}$ | $\mathbf{0.1032}_{\pm0.0006}$ | $\mathbf{0.0665}_{\pm0.0003}$ | $\mathbf{0.0687}_{\pm0.0001}$ | $\mathbf{0.0429}_{\pm0.0001}$ |
| | Improv. | 0.30% | 0.92% | 1.34% | 1.75% | 0.98% | 1.37% | 5.53% | 7.79% |
| | SSM | $0.3807_{\pm0.0015}$ | $0.3002_{\pm0.0012}$ | $0.3162_{\pm0.0027}$ | $0.2818_{\pm0.0023}$ | $0.1079_{\pm0.0005}$ | $0.0697_{\pm0.0002}$ | $0.0686_{\pm0.0002}$ | $0.0426_{\pm0.0001}$ |
| | NT-SSM | $\mathbf{0.4056}_{\pm0.0023}$ | $\mathbf{0.3173}_{\pm0.0015}$ | $\mathbf{0.3693}_{\pm0.0008}$ | $\mathbf{0.3389}_{\pm0.0007}$ | $\mathbf{0.1105}_{\pm0.0003}$ | $\mathbf{0.0711}_{\pm0.0003}$ | $\mathbf{0.0865}_{\pm0.0003}$ | $\mathbf{0.0539}_{\pm0.0002}$ |
| | Improv. | 6.54% | 5.70% | 16.79% | 20.26% | 2.41% | 2.01% | 26.09% | 26.53% |

