# OpenReview forum: "Rethinking Contrastive Learning for Graph Collaborative Filtering: Limitations and a Simple Remedy"
_ICML.cc/2026/Conference — ICML 2026 regular_

### Official Review · Reviewer_EnK1 · 2026-02-22

**Soundness:** 3
**Presentation:** 3
**Significance:** 3
**Originality:** 3
**Overall Recommendation:** 5
**Confidence:** 3

**Summary:**

This paper presents a principled analysis of contrastive learning in graph collaborative filtering (GCF). By unfolding the closed-form formulation of LightGCN (He et al., 2020), the authors show that user–item prediction scores are computed by aggregating interactions over a large number of multi-hop neighbor pairs. This perspective reveals that effective recommendation depends critically on selectively upweighting neighbor pairs whose members are structurally similar to the target user and item.
The paper revisits contrastive learning from the perspective of prediction-level learning dynamics rather than representation geometry alone. Based on their analysis, the authors propose NT-SSM, a neighbor type–aware contrastive objective designed to induce more desirable weight-update behavior in GCF.

**Compliance With Llm Reviewing Policy:**

Affirmed.

**Final Justification:**

I will maintain my original score

**Key Questions For Authors:**

1.	Why do gains under SimGCL remain relatively marginal compared to LightGCN?
Is this because augmentation-based objectives already partially compensate for SSM’s limitations?
2.	Can the authors analyze how optimal α values relate to dataset properties such as sparsity, clustering coefficient, or average degree?
3.	Is NT-SSM compatible with auxiliary contrastive objectives used in augmentation-based GCF methods (e.g., SGL, SimGCL)?
If so, how would the objectives interact?
4.	What is the computational overhead of NT-SSM compared to standard SSM?

**Limitations:**

It would improve readability to provide a concise, high-level overview of the proposed model earlier in the method section before presenting gradient derivations. Currently, the presentation becomes algebra-heavy before the intuition is fully consolidated.

**Strengths And Weaknesses:**

Strengths
•	The unfolding of LightGCN into a neighbor-pair aggregation perspective provides a clear and insightful reinterpretation of GCF’s prediction mechanism.
•	The shift from representation-level analysis (alignment/uniformity) to weight-update dynamics is novel and well justified.
•	The heuristic experiments (Figures 2 and 3) convincingly demonstrate that selective and type-dependent upweighting improves recommendation performance.
•	Experiments cover three GCF backbones across four datasets, with results averaged over five runs, indicating sound experimental protocol.
•	The ablation study (Table 2) clearly shows that each design component in NT-BPR and NT-SSM contributes to performance gains.

Weaknesses
1.	While NT-SSM shows strong improvements on LastFM and MovieLens, improvements on Amazon-Book and Yelp (especially under SimGCL) are relatively marginal. The paper does not sufficiently analyze why gains vary substantially across datasets and backbones. A deeper discussion grounded in dataset characteristics (e.g., sparsity, degree distribution, homophily strength) would strengthen the contribution.
2.	The paper does not analyze how training time scales with dataset size. Since NT-SSM introduces bidirectional objectives and type-specific modulation, it would be useful to clarify: Additional computational overhead compared to SSM or/ and Practical training-time impact
3.	Gains under SimGCL are relatively small. It remains unclear whether NT-SSM complements or partially overlaps with augmentation-based contrastive objectives.

---

> ### Author Rebuttal · Authors · 2026-03-31
>
> Dear Reviewer EnK1,
>
> We thank you for your insightful comments and careful evaluation. Below, we carefully address your feedback.
>
> ---
>
> # W1 + Q2. [Relation Between $\alpha$ and Dataset Properties]
>
> We thank the reviewer for the insightful feedback.
>
> - **[Optimal $\alpha$]** We report the selected $\alpha$ for each neighbor type.
> |  | $\alpha_U^{(U)}$ | $\alpha_I^{(I)}$ | $\alpha_U^{(I)}$ | $\alpha_I^{(U)}$ |
> | --- | --- | --- | --- | --- |
> | LastFM | 1.2 | 0.8 | 0.8 | 0.9 |
> | ML-1M | 1.2 | 0.8 | 0.8 | 1.0 |
> | Yelp | 1.2 | 1.2 | 1.1 | 1.2 |
> | Amazon | 1.2 | 1.1 | 1.1 | 1.1 |
> - **[Connection to dataset properties]** We examine how $\alpha$ relates to data characteristics.
>     - We measure scale, sparsity (density_log), and skewness (gini_user, gini_item) [1].
>     - In **large/sparse** datasets (Yelp, Amazon), all $\alpha$ values are similarly high. Many neighbor pairs are noisy due to scale and sparsity, so stricter filtering is needed across all types.
>     - In **denser/skewed** datasets (LastFM, ML-1M), $\alpha$ is smaller and type-dependent. High skewness leads to different thresholds due to uneven informativeness.
>
> |  | # Users | # Items | density_log | gini_user | gini_item |
> | --- | --- | --- | --- | --- | --- |
> | LastFM | 1885 | 13973 | -2.612 | 0.025 | 0.701 |
> | ML-1M | 6039 | 3583 | -1.564 | 0.514 | 0.658 |
> | Yelp | 31668 | 38038 | -3.042 | 0.394 | 0.533 |
> | Amazon | 52643 | 91369 | -3.412 | 0.442 | 0.492 |
>
> We will include this analysis and statistics in the revised paper.
>
> [1] The Datasets Dilemma: How Much Do We Really Know About Recommendation Datasets? (WSDM 2022)
>
> ---
>
> # W2 + Q4. [Training Time]
>
> As the reviewer suggested, we analyze the efficiency of NT-SSM compared to SSM. NT-SSM introduces only a modest computational overhead over SSM.
>
> - **[Graph propagation]** A single graph propagation takes $O(L|E|d)$ per epoch, where $L$ is the number of GCN layers, $|E|$ is the number of interactions, and $d$ is the embedding dimension. This cost is identical for NT-SSM and SSM, and thus the inference time remains the same.
> - **[Loss computation]** SSM costs $O(|E||B_u|d)$ per epoch. NT-SSM adds (1) a constant factor for computing neighbor type-decomposed inner products per negative sample, and (2) scoring $|B_i|$ negative users from the bidirectional objective. This results in $O(|E|(|B_u|+|B_i|)d)$ per epoch.
> - **[Empirical analysis]** We measure training time (sec/epoch) using RTX3090Ti GPU. As shown in the table below, NT-SSM introduces a modest overhead of 1.25-1.54x over SSM, which decreases on larger datasets, while yielding substantial accuracy gains (e.g., up to 21.45% NDCG@20 improvements in ML-1M).
>
> |  | LightGCN (SSM) | LightGCN (NT-SSM) |
> | --- | --- | --- |
> | LastFM | 0.471 | 0.727 |
> | ML-1M | 6.164 | 8.696 |
> | Yelp | 16.811 | 22.682 |
> | Amazon | 39.909 | 48.088 |
>
> ---
>
> # W3 + Q1 + Q3.  [Varying Gains Under SimGCL/NCL]
>
> As the reviewer mentioned, the performance gains on SimGCL (and NCL) are relatively smaller compared to those on LightGCN, and we provide further insights below.
>
> - **[Similarity]** Both approaches can be viewed as selectively upweighting neighbor pairs.
>     - SSM operates on user-item pairs $(u,i)$, selectively upweighting neighbor pairs $(v,v’)$ with $v\in N_u$ and $v’\in N_i$.
>     - SimGCL introduce contrastive signals on item-item (and/or user-user) pairs, selectively upweighting neighbor pairs $v,v’\in N_i$ (or $N_u$).
>     - Since SimGCL partially realize neighbor pair selection, NT-SSM's additional benefit is smaller but remains complementary.
> - **[Key distinctions]** NT-SSM is distinct in both scope and design.
>     - NT-SSM considers neighbor pairs between users and items, grounded in the GCF prediction mechanism that decomposes scores into UU/II/UI/IU types.
>     - SimGCL focuses on neighbors of the same node, and their CL losses do not align with GCF prediction mechansim.
> - **[Extension]** As the reviewer suggested, we extend existing CL objectives used in SimGCL by introducing type-specific $\alpha$ (i.e., NT-CL).
>     - As shown below, applying NT-CL to SimGCL improves over the original CL.
>     - But, NT-SSM is more effective than NT-CL, as it directly aligns with the GCF prediction mechanism.
>
> We will include this discussion in the revised manuscript.
>
> | NDCG@20 | SSM + CL | SSM + NT-CL | NT-SSM + CL |
> | --- | --- | --- | --- |
> | LastFM | 0.2562 | 0.2662 | **0.2726** |
> | ML-1M | 0.2899 | 0.3125 | **0.3258** |
>
> ---
>
> # L1. [Readability]
>
> We thank the reviewer for the helpful suggestion. We agree that providing a high-level overview before the derivations would improve readability.
>
> In the revised manuscript, we will:
>
> - add a concise overview of the proposed method earlier in the section.
> - include an illustrative figure to convey the intuition of neighbor-pair selection and type-aware upweighting.
> - clarify key notations before presenting the derivations.
>
> We will incorporate these improvements using the additional space available in the camera-ready version.

---

> > ### Author Rebuttal · Reviewer_EnK1 · 2026-04-02
> >
> > All major concerns are addressed. I maintain my Accept recommendation

---

> > > ### Author Response · Authors · 2026-04-05
> > >
> > > Dear Reviewer EnK1,
> > >
> > > Thank you very much for your valuable feedback and comments. We will carefully incorporate them to improve the final version of our paper.
> > >
> > > Best regards,\
> > > The Authors

---

### Official Review · Reviewer_9zCt · 2026-03-02

**Soundness:** 3
**Presentation:** 3
**Significance:** 3
**Originality:** 3
**Overall Recommendation:** 5
**Confidence:** 5

**Summary:**

This paper studies contrastive-learning-based optimization in graph collaborative filtering, focusing on Sampled Softmax (SSM). The authors analyze LightGCN by decomposing prediction scores into multi-hop neighbor-pair interactions, and argue that performance depends on which neighbor pairs are emphasized during training. Motivated by empirical findings, they propose NT-SSM, a neighbor-type-aware bidirectional extension of SSM, and show interesting improvements over SSM/BPR across multiple datasets and GCF backbones.

**Compliance With Llm Reviewing Policy:**

Affirmed.

**Final Justification:**

The authors addressed all my concerns.

**Key Questions For Authors:**

- Q1. The theoretical analysis and the Eq. (1)-style neighbor-pair decomposition appear to be developed under an inner-product scoring formulation. Could the authors clarify which parts of the analysis and the NT-SSM design are specific to dot-product/bilinear scoring, and which parts are expected to generalize to alternative scoring functions?

- Q2. Could the authors clarify how negative samples are constructed and used in all experiments (for both SSM and NT-SSM)? In addition, have the authors evaluated whether the observed “desirable learning dynamics” and the gains of NT-SSM persist under stronger hard-negative sampling strategies? If not, the authors are encouraged to explicitly discuss this.

- Q3. Could the authors provide training-cost comparisons between the proposed method and the main baselines?

**Limitations:**

The authors include a brief broader impact statement and appropriately mention popularity bias and privacy considerations in recommender systems. However, the limitations discussion would be stronger if it also explicitly acknowledged methodological limitations: (i) the analysis is largely instantiated under dot-product scoring, and may not directly transfer to metric-learning or other non-bilinear recommenders; (ii) the induced optimization dynamics may depend on the negative sampling policy; and (iii) NT-SSM introduces additional hyperparameter/tuning overhead relative to SSM/BPR.

**Strengths And Weaknesses:**

# Main Strengths
- The paper provides a clear and insightful view of GCF optimization by unfolding the prediction score into an aggregation over multi-hop neighbor-pair interactions.
- The empirical study on desirable learning dynamics is well motivated, which shows that selectively using structurally similar neighbors (instead of all neighbors) improves a proxy prediction score, and that the optimal retention behavior varies across neighbor pair types.
- It introduces a bidirectional objective NT-SSM with a corresponding gradient analysis showing how both user- and item-side structural similarities affect updates.

# Main Weaknesses
- The analysis and the neighbor-pair weighting interpretation appear to rely heavily on an inner-product scoring form (both in the prediction decomposition and in the gradient derivation w.r.t. the pairwise term $e_v^\top\cdot e_{v'}$)​. While the SSM definition initially allows generic similarity (e.g., inner product or cosine), the key derivation is instantiated with inner product “for simplicity.” It is therefore unclear how well the conclusions (and NT-SSM’s design rationale) transfer to non-dot-product recommenders, especially metric-learning-based formulations or other non-bilinear scorers [1,2,3].
- The performance gain is uneven across settings. In particular, some SSM→NT-SSM results are marginal (and even slightly negative in one Amazon-Book LightGCN setting in Table 1), so the practical impact should be discussed more carefully rather than summarized only as uniformly strong.
- The method introduces additional flexibility (bidirectional formulation + multiple type-specific coefficients) and the paper tunes four NT-SSM coefficients with Optuna, but does not clearly quantify the added computational/tuning overhead relative to SSM/BPR. Runtime, memory, and tuning-budget comparisons would be important for a fair practical assessment.
- The key update dynamics depend explicitly on sampled negative sets (e.g., $B_u$ and $B_i$ in NT-SSM) and the induced distributions (e.g., $\pi_u$)​. However, the paper does not sufficiently discuss sensitivity to the negative sampling strategy (uniform vs. popularity-biased vs. hard-negative sampling [4]). Since stronger samplers can alter the sampled distribution and optimization trajectory, the claimed “desirable learning dynamics” and NT-SSM gains may be sampling-policy-dependent.

# Reference
- [1] Collaborative Metric Learning

- [2] Rethinking collaborative metric learning: Toward an efficient alternative without negative sampling

- [3] The minority matters: A diversity-promoting collaborative metric learning algorithm

- [4] On the Theories Behind Hard Negative Sampling for Recommendation

---

> ### Author Rebuttal · Authors · 2026-03-31
>
> Dear Reviewer 9zCt,
>
> We are grateful for your careful evaluation and valuable feedback.
>
> ---
>
> # W1 + Q1 + L1. [Scoring Functions]
> - **[Why inner product]** We adopt the inner product in our analysis to obtain interpretable theoretical properties and tractable closed-form results, as it is the most commonly used in GCF.
> - **[What is specific]** Under the inner product, we derive (i) the closed-form prediction decomposition into neighbor-pair interactions (Eq. (1)) and (ii) the gradient of SSM (Eq. (5)), both relying on bilinearity.
> - **[What is general]** Our core insights are not tied to the inner product.
>     - Even with a general scorer, LightGCN produces embeddings as neighbor aggregations, so predictions depend on interactions between neighbor sets, regardless of the scoring function.
>     - Thus, interpreting GCF learning as selective neighbor pair reweighting does not depend on bilinearity. In Section 4, we use co-occurrence-based scoring (Eq. (4)), not the dot product.
>
> We will clarify this discussion in the revised manuscript.
>
> ---
>
> # W2. [Uneven Performance Gains]
> - **[Optimal $\alpha$]** We report the selected $\alpha$ for each neighbor type.
> |  | $\alpha_U^{(U)}$ | $\alpha_I^{(I)}$ | $\alpha_U^{(I)}$ | $\alpha_I^{(U)}$ |
> | --- | --- | --- | --- | --- |
> | LastFM | 1.2 | 0.8 | 0.8 | 0.9 |
> | ML-1M | 1.2 | 0.8 | 0.8 | 1.0 |
> | Yelp | 1.2 | 1.2 | 1.1 | 1.2 |
> | Amazon | 1.2 | 1.1 | 1.1 | 1.1 |
> - **[Connection to dataset properties]** We examine how $\alpha$ relates to data characteristics.
>     - We measure scale, sparsity (density_log), and skewness (gini_user, gini_item) [1].
>     - In **large/sparse** datasets (Yelp, Amazon), all $\alpha$ values are similarly high. Many neighbor pairs are noisy due to scale and sparsity, so stricter filtering is needed across all types.
>     - In **denser/skewed** datasets (LastFM, ML-1M), $\alpha$ is smaller and type-dependent. High skewness leads to different thresholds due to uneven informativeness.
>
> |  | # Users | # Items | density_log | gini_user | gini_item |
> | --- | --- | --- | --- | --- | --- |
> | LastFM | 1885 | 13973 | -2.612 | 0.025 | 0.701 |
> | ML-1M | 6039 | 3583 | -1.564 | 0.514 | 0.658 |
> | Yelp | 31668 | 38038 | -3.042 | 0.394 | 0.533 |
> | Amazon | 52643 | 91369 | -3.412 | 0.442 | 0.492 |
>
> We will include this analysis and statistics in the revised paper.
>
> [1] The Datasets Dilemma: How Much Do We Really Know About Recommendation Datasets? (WSDM 2022)
>
> ---
>
> # W3 + L3. [Computational overhead]
> - **[Computational efficiency]**
>     - **[Graph propagation]** It takes $O(L|E|d)$ per epoch ($L$ = \# layers, $|E|$ = \# interactions, $d$ = dimension). The inference time remains the same.
>     - **[Loss computation]** SSM costs $O(|E||B_u|d)$ per epoch. NT-SSM adds (1) neighbor type-decomposed inner products per negative sample, and (2) $|B_i|$ negative users from the bidirectional objective, resulting in $O(|E|(|B_u|+|B_i|)d)$ per epoch.
>     - **[Empirical analysis]** We measure training time (sec/epoch, RTX3090Ti) below. NT-SSM's overhead decreases on larger datasets, while achieving substantial gains (e.g., 21.45% NDCG@20 gain in ML-1M).
>
> | sec/epoch  | SSM | NT-SSM |
> | --- | --- | --- |
> | LastFM | 0.47 | 0.73 |
> | ML-1M | 6.16 | 8.70 |
> | Yelp | 16.81 | 22.68 |
> | Amazon | 39.91 | 50.05 |
> - **[Memory efficiency]** NT-SSM does not introduce additional learnable parameters; the bidirectional loss reuses the same batch. Peak GPU memory below shows marginal overhead.
>
> | MB | SSM | NT-SSM |
> | --- | --- | --- |
> | LastFM | 42.9 | 47.7 |
> | ML-1M | 54.7 | 56.9 |
> | Yelp | 163.0 | 179.2 |
> | Amazon | 295.4 | 329.9 |
>
> - **[Tuning]** We tune four $\alpha$s via Optuna with 100 trials per dataset (<1% of the $11^4$ grid).
>
> ---
>
> # W4 + Q2 + L2. [Negative sampling]
> - **[Current strategy]** We uniformly sample user-item interaction pairs and use in-batch negatives, implicitly sampling negatives proportional to their frequency.
> - **[Effect]** The sampling strategy determines the distribution $\pi_u$ in Eq. (5), affecting the upweighting threshold and thus the optimal $\alpha$ values.
> - **[Experiments]** We examine two alternatives: **DNS** (hard negatives of size $m$; default $m=2048$ batch size) and popularity-biased sampling ($\propto (d_u + d_i)^\gamma$ where $d_v$ = degree of node $v$; default $\gamma=0$). As shown below, the default performs best, and optimal $\alpha$ varies across strategies, consistent with our analysis above.
>
> |  |  | Default | DNS (m=256) | DNS (m=1024) | pop (γ=-1.0) | pop (γ=1.0) |
> |---|---|---|---|---|---|---|
> | LastFM | NDCG@20 | **0.2709** | 0.2303 | 0.2620 | 0.2322 | 0.2451 |
> |  | α_U^U, α_I^I, α_U^I, α_I^U | 1.2 / 0.8 / 0.8 / 0.9 | 0.8 / 0.8 / 0.8 / 0.8 | 1.2 / 1.2 / 1.2 / 0.8 | 1.0 / 0.8 / 0.8 / 1.0 | 0.8 / 0.8 / 0.8 / 0.8 |
> | ML-1M | NDCG@20 | **0.3216** | 0.1928 | 0.3120 | 0.2968 | 0.2873 |
> |  | α_U^U, α_I^I, α_U^I, α_I^U | 1.2 / 0.8 / 0.8 / 1.0 | 1.2 / 1.0 / 1.0 / 1.2 | 1.2 / 0.8 / 0.8 / 1.2 | 1.0 / 0.8 / 1.0 / 1.0 | 0.8 / 1.0 / 0.8 / 1.0 |

---

> > ### Author Rebuttal · Reviewer_9zCt · 2026-04-01
> >
> > Thank you for your efforts. After reviewing your response, I still have concerns about the generalizability of the conclusions, as well as the design rationale of NT-SSM, particularly with respect to metric-learning-based formulations and other non-bilinear scorers. It would be preferable for the authors to provide a more thorough and sufficient discussion of these issues.

---

> > > ### Author Response · Authors · 2026-04-05
> > >
> > > Dear Reviewer 9zCt,
> > >
> > > We deeply appreciate the reviewer for this insightful question.
> > >
> > > ---
> > >
> > > ## [Extension to Non-Bilinear Scoring Functions]
> > >
> > > We clarify how our findings translate to L2 distance, which is a representative non-bilinear scoring function.
> > >
> > > - **[Expansion under non-linear scoring]** We consider the negative L2 distance as a representative non-bilinear scoring function:
> > >
> > >   $-d(u,i)=-\\|e\_u-e\_i\\|\_2^2$
> > >
> > >   Under LightGCN’s linear message passing, the final embeddings are linear aggregations of multi-hop neighbors’ initial embeddings:
> > >
> > >   $e_u=\\sum\_{v\in\tilde{N}_u} \\tilde{S}\_{uv} e_v^{(0)}$
> > >   $e_i=\\sum\_{v'\in\tilde{N}_i} \\tilde{S}\_{iv'} e\_{v'}^{(0)}$
> > >
> > >   If we expand the negative squared L2 distance between $e_u$ and $e_i$, we obtain:
> > >
> > >   $-d(u,i)$
> > >
> > >   $=2e_u^Te_i - \|e_u\|^2 - \|e_i\|^2$
> > >
> > >   $=2\sum\_{v\in \tilde{N}\_u}\sum\_{v'\in\tilde{N}\_i}\tilde{S}\_{uv}\tilde{S}\_{iv'}(e_v^{(0)T}e_{v'}^{(0)}) - \sum\_{v\in \tilde{N}\_u}\sum\_{v''\in\tilde{N}\_u}\tilde{S}\_{uv}\tilde{S}\_{uv''}(e_v^{(0)T}e_{v''}^{(0)})- \sum\_{v'\in \tilde{N}\_i}\sum\_{v'''\in\tilde{N}\_i}\tilde{S}\_{iv'}\tilde{S}\_{iv'''}(e_{v'}^{(0)T}e_{v'''}^{(0)})$
> > >
> > >   This expansion yields three components: (1) interactions between multi-hop neighbors of the user and the item, (2) interactions among the user’s multi-hop neighbors, and (3) interactions among the item’s multi-hop neighbors.
> > >
> > >   Importantly, the core mechanism of GCF learning, interpreted as weighting neighbor-pair interactions, remains unchanged. Compared to the bilinear scoring, this non-bilinear scoring additionally incorporates interactions among neighbors of users and items.
> > >
> > > - **[Extension of NT-SSM]** We extend NT-SSM to this setting by replacing inner-product with the negative L2 distance.
> > >
> > >   $\mathcal{L}(i;u)=-\log\frac{\exp(-d(u,i)/\tau)}{\exp(-d(u,i)/\tau)+\sum_{j\in B_u}\exp(-\tilde{d}(u,j)/\tau)}$
> > >
> > >   where $\tilde{d}(u,j)$ is defined as:
> > >
> > >   $\tilde{d}(u,j)=\|e_u\|_2^2 + \|e_j\|_2^2 - 2\left( \alpha_I^{(U)} s_I^{(U)}(u,j)+\alpha_I^{(I)}s_I^{(I)}(u,j) \right)$
> > >
> > >   where $s_I^{(t)}(u,j)$ is the partial similarity contributed by neighbor pairs involving item j’s neighbors of type $t\in\\{U,I\\}$, i.e., $s\_I^{(t)}(u,j)=\sum\_{v\in \tilde{N}\_u}\sum\_{v'\in\tilde{N}\_j^{(t)}} \tilde{S}\_{uv}\cdot \tilde{S}\_{jv'}\cdot (e_v^{(0)\top}e_{v'}^{(0)})$.
> > >
> > >   This formulation applies type-specific scaling to cross neighbor-pair interactions in the negative terms.
> > >
> > >   Empirically, NT-SSM outperforms the naive SSM with negative L2 distance (e.g., 29.9% gain in ML-1M), suggesting that modeling neighbor types leads to more effective learning dynamics with a non-bilinear scoring function.
> > >
> > >   |  | Non-bilinear (SSM) | Non-bilinear (NT-SSM) |
> > >   | --- | --- | --- |
> > >   | LastFM | 0.2399 | **0.2436** |
> > >   | ML-1M | 0.2271 | **0.2950** |
> > >
> > > ---
> > >
> > > ## [Extension to Metric Learning Formulations]
> > >
> > > As the reviewer suggested, we extended our framework to a metric learning formulation. We adapted CML [1] by integrating our type-aware neighbor upweighting dynamics into its distance-based objective (i.e., NT-CML).
> > >
> > > Specifically, we consider the standard CML loss:
> > >
> > > $\mathcal{L}\_{CML} = \max(0, d(u,i) - d(u,j) + \gamma)$
> > >
> > > where $d(u,i)=\|e_u-e_i\|_2^2=\|e_u\|_2^2+\|e_i\|_2^2-2e_u^\top e_i$.
> > >
> > > We extend this to NT-CML by replacing the standard negative distance calculation with a neighbor-type-aware distance:
> > >
> > > $\mathcal{L}_{NT-CML}=\max(0, d(u,i)-\tilde{d}(u,j)+\gamma)$
> > >
> > > where $\tilde{d}(u,j)$ is defined as:
> > >
> > > $\tilde{d}(u,j)=\|e_u\|_2^2 + \|e_j\|_2^2 - 2\left( \alpha_I^{(U)} s_I^{(U)}(u,j)+\alpha_I^{(I)}s_I^{(I)}(u,j) \right)$
> > >
> > > where $s_I^{(t)}(u,j)$ is the partial similarity contributed by neighbor pairs involving item j’s neighbors of type $t\in\\{U,I\\}$, i.e., $s\_I^{(t)}(u,j)=\sum\_{v\in \tilde{N}\_u}\sum\_{v'\in\tilde{N}\_j^{(t)}} \tilde{S}\_{uv}\cdot \tilde{S}\_{jv'}\cdot (e_v^{(0)\top}e_{v'}^{(0)})$.
> > >
> > > That is, by applying our neighbor-type-aware scaling coefficients to the cross-term of the distance metric, we adopt distinct weights on the negative samples with respect to their types. Notably, when $\alpha=1$ for all coefficients, it is equivalent to the original CML objective.
> > >
> > > We empirically validate this extension, demonstrating the effectiveness of NT-CML:
> > >
> > > |  | CML | NT-CML |
> > > | --- | --- | --- |
> > > | LastFM | 0.1891 | **0.2125** |
> > > | ML-1M | 0.1780 | **0.2725** |
> > >
> > > [1] Collaborative Metric Learning (WWW 2017)
> > >
> > > ---
> > >
> > > We acknowledge that our analysis may not readily generalize to complex scoring functions composed of multiple non-linear transformations (e.g., MLP-based scorers), where such a structural breakdown becomes less explicit. We will clarify this scope and limitation of our theoretical unfolding in the revised manuscript.

---

### Official Review · Reviewer_8g7i · 2026-03-13

**Soundness:** 3
**Presentation:** 4
**Significance:** 2
**Originality:** 3
**Overall Recommendation:** 3
**Confidence:** 4

**Summary:**

This paper starts from the impact of contrastive learning on the prediction mechanism of GCF, deconstructing the LightGCN model from a novel "learning dynamics" perspective. It analyzes that the SSM loss centers its weight updates around item-side neighbors. The authors propose the NT-loss, which incorporating a bidirectional loss and a type-aware weighted optimization objective. Experimental results shows significant improvements across multiple datasets and base models.

**Compliance With Llm Reviewing Policy:**

Affirmed.

**Final Justification:**

The rebuttal usefully addressed several of my concerns, especially through added efficiency analysis, broader experiments, and more direct empirical evidence for the claimed learning dynamics. However, I still think the validation of these dynamics is not yet fully sufficient, and the analysis is most clearly supported under LightGCN’s linear aggregation setting, so I maintain my original scores and Weak Reject recommendation.

**Key Questions For Authors:**

1. How do SSM and NT-SSM compare in parameters and computational efficiency?
2. How does NT-SSM concretely learn better optimization dynamics during training? Is there any visualization or metric to support?
3. Under the same performing setting, what is the distribution of these type-dependent weights?

**Limitations:**

The authors need to discuss the limitations of their method under linear aggregation settings.

**Strengths And Weaknesses:**

Strengths:

1. The paper is well written and has a clear narrative. Its overall storyline, from theoretical analysis to empirical improvement, is easy to follow.
2. The analytical perspective shows originality. Rethinking GCF prediction as neighbor-pair aggregation, and further revisiting SSM from the perspective of pair-weight update dynamics, is a valuable viewpoint.
3. The experimental results shows effectiveness and provide solid support for the theoretical analysis.

Weaknesses:

1. Soundness：The paper’s claim is that NT-SSM learns more desirable neighbor-pair dynamics. However, besides recommendation performance metrics, the paper does not provide direct empirical evidence, such as visualizations of whether these dynamics improve or not, lack of empirical validation.
2. Significance: The analysis is centered mainly on LightGCN, and the linear structure itself is an important premise underlying the proposed perspective. Applicability to methods with more complex aggregation mechanisms, such as Transformer-based GCF models, appears unknown.
3. Originality: As an analysis-driven improvement, NT-SSM is reasonable, but from the perspective of algorithm design, its novelty is somewhat limited. In particular, the bidirectional contrastive loss is already a standard technique in multi-view learning.

---

> ### Author Rebuttal · Authors · 2026-03-31
>
> Dear Reviewer 8g7i,
>
> We deeply appreciate your detailed review and helpful suggestions.
>
> ---
>
> # W1 + Q2. [Empirical Evidence]
>
> We thank the reviewer for highlighting this important aspect.
>
> - **[Metric]** For each type (UU/II/UI/IU), we split neighbor pairs into informative (Info.) and uninformative (Uninfo.) ones using the optimal retention ratio (Figure 3), and compare their average embedding dot products. Desirably, informative pairs should exhibit higher values with a clear gap.
> - **[Empirical evidence]** As shown below, both SSM and NT-SSM yield higher dot products for informative pairs, with NT-SSM showing a larger gap, indicating more effective upweighting of informative pairs.
>
> |  |  | UU | II | UI | IU |
> | --- | --- | --- | --- | --- | --- |
> | SSM | Info.| 0.0040 (x2.67) | 0.0106 (x1.16) | 0.0062 (x1.77) | 0.0053 (x1.83) |
> |  | Uninfo. | 0.0015 | 0.0091  | 0.0035 | 0.0029 |
> | NT-SSM | Info. | 0.0775 (**x2.80**) | 0.0441 (**x5.80**) | 0.1382 (**x81.29**) | 0.0056 (**x4.00**) |
> |  | Uninfo. | 0.0277 | 0.0076 | 0.0017 | 0.0014 |
>
> ---
>
> # **W2 + L1.** [Other GCF Models]
> - **[Experiments]** We evaluate NT-SSM on GCF models with more cmoplex aggregation, including NGCF [1] and GFormer [2]. As shown below, NT-SSM improves over SSM across these models, demonstrating its applicability.
> - **[Why LightGCN]**
>     - Despite the applicability of NT-SSM, we adopt LightGCN due to its simplicity and analytical tractability. It is widely adopted (+6K citations) across domains (e.g., social, bundle, multimedia recommendations).
>     - As suggested, we will strengthen the limitation section by acknowledging that our analysis and method are developed under LightGCN's linear aggregation framework.
>
> | NDCG@20 | NGCF (SSM) | NGCF (NT-SSM) | GFormer (SSM) | GFormer (NT-SSM) |
> |---|---|---|---|---|
> | LastFM | 0.2563 | **0.2607** | 0.1609 | **0.2496** |
> | ML-1M  | 0.2093 | **0.2946** | 0.2447 | **0.3071** |
>
> [1] Neural Graph Collaborative Filtering (SIGIR 2019)\
> [2] Graph Transformer for Recommendation (SIGIR 2023)
>
> ---
>
> # W3. [Technical Novelty]
>
> We believe our work provides technically meaningful contributions.
>
> - **[1. Neighbor pair type-aware design]**
>     - We propose the first CL objective that explicitly models and controls neighbor-pair types through type-specific upweighting thresholds.
>     - We further validate this by measuring performance by neighbor pair type; NT-SSM improves SSM across all types.
> | NDCG@20 | UU | II | UI | IU |
> | --- | --- | --- | --- | --- |
> | SSM | 0.2841 | 0.1246 | 0.0920 | 0.2818 |
> | NT-SSM | **0.3101** | **0.3099** | **0.3139** | **0.2993** |
> - **[2. Principled bidirectional design]**
>     - We introduce a bidirectional objective performing neighbor selection on both user and item sides.
>     - This is not simply borrowed but derived from our analysis that selection from both sides is crucial (Section 4).
>     - We empirically validate this through ablation studies (Section 7.3).
>
> All design choices are grounded in our analysis.
>
> - **[New perspective]** Prior CL for GCF (e.g., SimGCL) focuses on embedding geometry (alignment, uniformity), while we introduce an optimization perspective interpreting GCF learning as selective neighbor pair upweighting (Section 3).
> - **[Grounded design]** NT-SSM is grounded in empirical (Section 4) and theoretical (Section 5) analysis, aligned with the identified desirable learning dynamics.
>
> ---
>
> # Q1. [Efficiency]
> - **[Parameter efficiency]** NT-SSM introduces no additional learnable parameters compared to SSM.
> - **[Computational efficiency]**
>     - **[Graph propagation]** It takes $O(L|E|d)$ per epoch ($L$ = \# of GCN layers, $|E|$ = \# of interactions, $d$ = dimension). Thus, the inference time remains the same.
>     - **[Loss computation]** SSM costs $O(|E||B_u|d)$ per epoch. NT-SSM adds (1) neighbor type-decomposed inner products per negative sample, and (2) $|B_i|$ negative users from the bidirectional objective, resulting in $O(|E|(|B_u|+|B_i|)d)$ per epoch.
>     - **[Empirical analysis]** We measure training time (sec/epoch, RTX3090Ti) below. NT-SSM's overhead decreases on larger datasets, while achieving substantial gains (e.g., 21.45% NDCG@20 improvements in ML-1M).
>
> | sec/epoch  | SSM | NT-SSM |
> | --- | --- | --- |
> | LastFM | 0.471 | 0.727 |
> | ML-1M | 6.164 | 8.696 |
> | Yelp | 16.811 | 22.682 |
> | Amazon | 39.909 | 50.049 |
>
> ---
>
> # Q3. [Optimal $\alpha$]
> - **[Hyperparameters]** We report NT-SSM hyperparameters below. We will include NT-BPR hyperparameters in the revised manuscript.
> - **[Implication]** The optimal $\alpha$ values vary across datasets and neighbor types, supporting type-specific weighting (Figure 3).
>
> |  | $\alpha_U^{(U)}$ | $\alpha_I^{(I)}$ | $\alpha_U^{(I)}$ | $\alpha_I^{(U)}$ |
> | --- | --- | --- | --- | --- |
> | LastFM | 1.2 | 0.8 | 0.8 | 0.9 |
> | ML-1M | 1.2 | 0.8 | 0.8 | 1.0 |
> | Yelp | 1.2 | 1.2 | 1.1 | 1.2 |
> | Amazon | 1.2 | 1.1 | 1.1 | 1.1 |

---

> > ### Author Rebuttal · Reviewer_8g7i · 2026-04-04
> >
> > Thank you for the detailed rebuttal. My concerns are partially resolved. The added efficiency analysis and the additional results on NGCF/GFormer are helpful, and the informative/uninformative pair statistics provide some empirical support for the claimed learning dynamics. However, this evidence is still somewhat indirect relative to my original concern about direct validation of the optimization dynamics, so I encourage the authors to provide clearer evidence in the final version if possible, and to more explicitly discuss which parts of the analysis rely on LightGCN’s linear aggregation setting.

---

> > > ### Author Response · Authors · 2026-04-05
> > >
> > > Dear Reviewer 8g7i,
> > >
> > > We sincerely thank the reviewer for this insightful question.
> > >
> > > ---
> > >
> > > ## [More Direct Empirical Evidence]
> > >
> > > To directly validate that NT-SSM induces the desired optimization dynamics, we compare the gradient dynamics between important and unimportant neighbor pairs during training.
> > >
> > > - **[Analysis setup]** We measure the gradient magnitude assigned to each neighbor pair, i.e., $|\frac{\partial \mathcal{L}}{\partial (e_v^{(0)\top} e_{v'}^{(0)})}|$. For each neighbor type (UU/II/UI/IU), we split the neighbor pairs into informative and uninformtive ones based on their structural similarity, using the empirically best retention ratios identified in Figure 3. Then, we compare the average gradient magnitude assigned to each subset, averaged over the first 10 epochs during training. As training encourages informative (or positive) pairs to have higher similarity, larger gradients correspond to stronger updates that amplify this effect.
> > > - **[Empirical results]** As shown below, NT-SSM more effectively applies stronger gradients to informative neighbor pairs than to uninformative ones in MovieLens. This suggests that NT-SSM induces more desirable learning dynamics, closely aligning with our empirical analysis in Section 4.
> > >
> > > |  |  | UU | II | UI | IU |
> > > | --- | --- | --- | --- | --- | --- |
> > > | SSM | Informative | 0.0072 (x1.07) | 0.0085 (x0.98) | 0.0090 (x1.11) | 0.0072 (x1.06) |
> > > |  | Uniformative | 0.0067 | 0.0087 | 0.0081 | 0.0068 |
> > > | NT-SSM | Informative | 0.0121 (**x1.29**) | 0.0085 (**x1.47**) | 0.0174 (**x2.20**) | 0.0069 (**x1.08**) |
> > > |  | Uninformative | 0.0094 | 0.0058 | 0.0079 | 0.0064 |
> > > - **[Connection to performance]** The improvement in learning dynamics is reflected in performance gains. We measure performance separately for each neighbor pair type by restricting the prediction using only specific neighbor types (e.g., for UU, we use $\tilde{N}_u^{(U)}$ and $\tilde{N}_i^{(U)}$). As shown below, NT-SSM outperforms SSM across all neighbor pair types. Notably, the performance gap is largest for the II and UI neighbor pairs, which aligns with the particularly large gradient magnitude gaps observed above between SSM and NT-SSM.
> > >
> > > | NDCG@20 | UU | II | UI | IU |
> > > | --- | --- | --- | --- | --- |
> > > | SSM | 0.2841 | 0.1246 | 0.0920 | 0.2818 |
> > > | NT-SSM | **0.3101** | **0.3099** | **0.3139** | **0.2993** |
> > >
> > > ---
> > >
> > > ## [Discussion on Linear Aggregation]
> > >
> > > We clarify which components of our analysis rely on linear aggregation and which aspects conceptually generalize beyond it.
> > >
> > > - **[What relies on linear aggregation]** Our exact mathematical derivations depend on the bi-linearity of the LightGCN formulation. Linear aggregation enables an additive decomposition of the prediction score into neighbor-pair interactions and allows us to isolate type-specific contributions and derive type-aware gradients and thresholds.
> > >     - In Eq. (1), the prediction score can be explicitly unfolded into an additive sum of multi-hop neighbor pair interactions, which relies on the distributive property of linear aggregation.
> > >     - In Eq. (5), the closed-form gradient with respect to pairwise interactions, as well as the upweighting thresholds derived from it, are formulated on this additive decomposition.
> > > - **[What generalizes beyond linear aggregation]** Our core conceptual insight, that GCF learning can be interpreted as selectively upweighting interactions between neighbor pairs, does not depend on linearity. Given a general, potentially non-linear transformation and aggregation function $g$, the final embeddings are still constructed from the initial embeddings of multi-hop neighbor sets:
> > >
> > >     $e_u=g(\\{e_x^{(0)}\mid x\in \tilde{N}_u\\})$  $e_i=g(\\{e_x^{(0)}\mid x\in \tilde{N}_i\\})$
> > >
> > >     These multi-hop neighbor sets can be decomposed based on their types (i.e., users and items):
> > >
> > >     $e_u=g(\\{e_{x'}^{(0)} \mid x'\in \tilde{N}_u^{(U)}\cup \tilde{N}_u^{(I)}\\})$
> > >
> > >     $e_i=g(\\{e_{x'}^{(0)} \mid x'\in \tilde{N}_i^{(U)}\cup\tilde{N}_i^{(I)}\\})$
> > >
> > >     This induces interactions across all neighbor-type pairs:
> > >
> > >     $\\{\\tilde{N}_u^{(U)}, \\tilde{N}_u^{(I)}\\} \\times \\{\\tilde{N}_i^{(U)}, \\tilde{N}_i^{(I)}\\}$
> > >
> > >     While explicit additive decomposition may not hold under non-linear $g$, the model still aggregates information across neighbor-type pairs, and training implicitly adjusts their relative influence via gradients.

---

### Official Review · Reviewer_SE5X · 2026-03-13

**Soundness:** 3
**Presentation:** 3
**Significance:** 3
**Originality:** 3
**Overall Recommendation:** 4
**Confidence:** 4

**Summary:**

From a high-level perspective, this paper is organized into four main parts:
1. **Problem formulation.** The authors investigate how contrastive learning influences the prediction performance of graph-based collaborative filtering, specifically focusing on the graph convolution process in LightGCN.

2. **Empirical observations.** Through a forward-pass analysis of graph convolution, the authors identify two empirical findings:
   (1) For a given user–item pair, only a small subset of neighbor pairs with high structural similarity significantly contribute to the final recommendation score. Selectively upweighting these influential pairs during graph convolution can enhance recommendation performance.
   (2) Neighbor pairs can be categorized into four types (i.e., U–U, I–I, U–I, and I–U), and the optimal structural similarity threshold for selection, as identified in observation (1), varies across these types.
These findings are clearly presented and well supported by empirical evidence.

3. **Proposed solution.** The authors analyze the gradient-based update dynamics of the SSM loss and identify certain limitations that prevent it from aligning with the desirable update behavior suggested by the empirical findings. In response, they propose a simple yet principled modification to the SSM loss, which partially addresses these issues.

4. **Experimental evaluation.** Experimental results demonstrate that the proposed loss function leads to improvements in recommendation performance.

Overall, this is a strong paper. It presents a rigorous analytical framework, supported by comprehensive empirical results and theoretical proofs. The writing is clear and accessible, and the findings are novel and insightful. The proposed solution is well aligned with the empirical observations. One minor limitation lies in the experimental evaluation section, which could be further strengthened; more detailed comments are provided in the strengths and weaknesses section.

**Compliance With Llm Reviewing Policy:**

Affirmed.

**Final Justification:**

The authors address my concerns, and I keep my original score.

**Key Questions For Authors:**

1. In Figure 3, the optimal retention ratios for different neighbor pair types exhibit substantial variation, ranging from 0.1% to 10%. It is worth discussing whether the four parameters $\alpha_{U}^{(U)}$, $\alpha_{U}^{(I)}$, $\alpha_{I}^{(U)}$, and $\alpha_{I}^{(I)}$ are sufficiently flexible to accommodate such a wide range of retention ratio values.
2. In Section 7.1, the datasets Yelp2018 and Amazon-Book are two traditional benchmarks also used in LightGCN and SimGCL. However, as shown in Table 3, the interaction volume of Amazon-Book differs slightly from that used in LightGCN and SimGCL. Moreover, in Table 1, the performance of LightGCN-BPR and LightGCN-SSM deviates from the results reported in their original papers. This discrepancy may stem from differences in data splitting strategies, and an explanation would help clarify the situation.
3. After running the provided code with the default parameter configuration of LightGCN-NT-SSM on the MovieLens dataset, I was unable to reproduce the performance reported in Table 2 (i.e., NDCG@20 = 0.2677). To facilitate reproducibility, could the optimal parameter settings be included in the code repository?

**Limitations:**

See Weaknesses.

**Strengths And Weaknesses:**

Strengths:
1. The findings presented in this paper are both novel and informative, aligning well with both intuitive reasoning and empirical observations. The experimental results in Section 4 are robust and persuasive, and the accompanying gradient analysis is theoretically sound.
2. The paper is clearly structured and well written, making it easy to understand for researchers in the field of graph-based collaborative filtering.
3. This work holds significant value for future research. The insights offered open up promising directions, such as designing more effective graph convolution mechanisms or developing unified frameworks that integrate graph collaborative filtering with contrastive learning.
4. Although the idea of emphasizing structurally similar neighbors in graph convolution may seem intuitive at first glance, this paper provides a rigorous and in-depth analysis that substantiates the approach. As such, the originality of the work is unquestionable.

Weaknesses:
1. In Section 6.1, providing a pseudo-code for the similarity calculation s(u,j) would help readers better understand the implementation details.
2. In Sections 7.1 and 7.4, the key hyperparameters of NT-SSM are $\alpha_{U}^{(U)}$, $\alpha_{U}^{(I)}$, $\alpha_{I}^{(U)}$, and $\alpha_{I}^{(I)}$. Although Figure 4 presents the sensitivity analysis for each hyperparameter, providing a recommended optimal parameter setting would facilitate better reproducibility.
3. In Table 1, the performance improvement of SimGCL and NCL is less significant than that observed on LightGCN. Given that SimGCL and NCL already incorporate additional contrastive learning components, a discussion on how the proposed method could be extended to such frameworks would provide valuable insights for future work.

---

> ### Author Rebuttal · Authors · 2026-03-30
>
> Dear Reviewer SE5X,
>
> We thank the reviewer for the careful evaluation and constructive feedback.
>
> ---
>
> # W1. [Implementation Details]
>
> We clarify our implementation as follows:
>
> - **[Graph propagation]** We follow LightGCN and separately aggregate embeddings from even and odd layers to capture multi-hop neighbors of users and items.
> - **[Loss computation]** We compute similarity (Eq. (6)) following standard SSM, but (1) apply neighbor type-specific $\alpha$ separately to similarities involving negative samples depending on their types, and (2) adopt a bidirectional loss with both user and item negatives.
> - **[Pseudo-code]** We will provide pseudo-code in Section 6.1 describing the above procedure.
>
> ---
>
> # W2 + Q3. [Optimal Hyperparameter Settings]
>
> We improve reproducibility as follows:
>
> - **[Hyperparameter settings]** We report the selected NT-SSM hyperparameters below. We will also include NT-BPR hyperparameters in the revised manuscript.
> - **[Reproducibility]** Based on the reviewer’s suggestion, we will update our codebase as follows:
>     - The current run.sh in the anoymous github sets all $\alpha$ to 1.0 by default. We will provide data-specific scripts with the selected optimal hyperparameters.
>     - We identified that arguments --alpha_uu and --alpha_ui are not included in the current run.sh (though used in our experiments). We will update the script to include them.
>
> |  | $\alpha_U^{(U)}$ | $\alpha_I^{(I)}$ | $\alpha_U^{(I)}$ | $\alpha_I^{(U)}$ | $\tau$ |
> | --- | --- | --- | --- | --- | --- |
> | LastFM | 1.2 | 0.8 | 0.8 | 0.9 | 0.2 |
> | ML-1M | 1.2 | 0.8 | 0.8 | 1.0 | 0.2 |
> | Yelp | 1.2 | 1.2 | 1.1 | 1.2 | 0.2 |
> | Amazon | 1.2 | 1.1 | 1.1 | 1.1 | 0.1 |
>
> ---
>
> # W3. [SimGCL/NCL Gains and Extension]
>
> As the reviewer mentioned, the performance gains on SimGCL (and NCL) are relatively smaller compared to those on LightGCN, and we provide further insights below.
>
> - **[Similarity]** Both approaches can be viewed as selectively upweighting neighbor pairs.
>     - SSM operates on user-item pairs $(u,i)$, selectively upweighting neighbor pairs $(v,v’)$ with $v\in N_u$ and $v’\in N_i$.
>     - SimGCL introduce contrastive signals on item-item (and/or user-user) pairs, selectively upweighting neighbor pairs $v,v’\in N_i$ (or $N_u$).
>     - Since SimGCL partially realize neighbor pair selection, NT-SSM's additional benefit is smaller but remains complementary.
> - **[Key distinctions]** NT-SSM is distinct in both scope and design.
>     - NT-SSM considers neighbor pairs between users and items, grounded in the GCF prediction mechanism that decomposes scores into UU/II/UI/IU types.
>     - SimGCL focuses on neighbors of the same node, and their CL losses do not align with GCF prediction mechansim.
> - **[Extension]** As the reviewer suggested, we extend existing CL objectives used in SimGCL by introducing type-specific $\alpha$ (i.e., NT-CL).
>     - As shown below, applying NT-CL to SimGCL improves over the original CL.
>     - But, NT-SSM is more effective than NT-CL, as it directly aligns with the GCF prediction mechanism.
>
> We will include this discussion in the revised manuscript.
>
> | NDCG@20 | SSM + CL | SSM + NT-CL | NT-SSM + CL |
> | --- | --- | --- | --- |
> | LastFM | 0.2562 | 0.2662 | **0.2726** |
> | ML-1M | 0.2899 | 0.3125 | **0.3258** |
>
> ---
>
> # Q1. [Flexibility of $\alpha$]
>
> We clarify the role and flexibilty of $\alpha$:
>
> - **[Role]** Each $\alpha$ controls the upweighting threshold in the gradient (Section 6.2); a neighbor pair is upweighted only when its structural similarity exceeds a threshold scaled by $\alpha$.
> - **[Connection to retention ratio]** This mechanism corresponds to controlling an effective retention ratio (Section 4). Smaller $\alpha$ retains more neighbor pairs; larger $\alpha$ applies stricter selection.
> - **[Flexibility]** When $\alpha=0$, all neighbor pairs are upweighted; when $\alpha\geq \tilde{\mathbf{S}}_{i}/\mathbb{E}_j [\tilde{\mathbf{S}}_j]$, no pairs are upweighted, showing that $\alpha$ flexibly controls a wide range of retention ratios.
>
> ---
>
> # Q2. [Dataset Preprocessing]
>
> We clarify our data preprocessing and implementation.
>
> - **[Dataset source]** We use the datasets from LightGCN++ [1], with 7:1:2 train/val/test splits.
> - **[Implementation]** Our implementation is based on SELFRec (+600 Github stars), which may introduce some differences (e.g., initialization) compared to the original implementations.
> - **[Experiments]** We conducted experiments on the Amazon-Book dataset from [2] for fair comparison. As shown below, NT-SSM (NT-BPR) improves over SSM (BPR) on both versions. We will include these results in the revised manuscript.
>
> | NDCG@20 | BPR | NT-BPR | SSM | NT-SSM |
> |---|---|---|---|---|
> | [1] (Used) | 0.0257 | 0.0284 | 0.0385 | 0.0386 |
> | [2] | 0.0260 | 0.0301 | 0.0412 | 0.0419 |
>
> [1] Revisiting LightGCN: Unexpected Inflexibility, Inconsistency, and a Remedy Towards Improved Recommendation (RecSys 2024)\
> [2] Neural Graph Collaborative Filtering (SIGIR 2019)

---

> > ### Author Rebuttal · Reviewer_SE5X · 2026-04-04
> >
> > Thank you for the authors’ detailed response, which addresses part of my concerns. The clarifications on implementation details and reproducibility, as well as the additional experiments and discussions, are helpful and improve the overall clarity of the paper. While some concerns remain partially addressed, particularly regarding the empirical gains on stronger baselines, I will maintain my original score.

---

> > > ### Author Response · Authors · 2026-04-05
> > >
> > > Dear Reviewer SE5X,
> > >
> > > Thank you for your thoughtful feedback and for acknowledging the improvements in clarity, reproducibility, and the additional experiments. We will further clarify and strengthen our discussion regarding the empirical gains in the final version.
> > >
> > > Best regards,\
> > > The Authors

---

### Decision · Program_Chairs · 2026-04-30

**Decision:**

Accept (regular)

**Comment:**

This paper revisits contrastive learning for graph collaborative filtering by analyzing user-item prediction as an aggregation over multi-hop neighbor-pair interactions and proposing a remedy that selectively emphasizes structurally similar neighbor pairs. Reviewers generally agreed that this perspective is insightful and that the paper offers a meaningful reinterpretation of optimization dynamics in graph collaborative filtering. The proposed objective is supported by strong results across datasets and backbones, and several reviewers emphasized that the work could influence future model and objective design in this area.

The main concern was whether the claimed learning dynamics are validated directly enough, and whether some parts of the analysis rely too strongly on the LightGCN-style setting or an inner-product formulation. One reviewer remained somewhat skeptical even after rebuttal, mainly because the direct empirical support for the intended dynamics could still be stronger.

That said, the rebuttal appears to have addressed many concerns, and the overall discussion remained clearly positive. In my view, the paper combines useful conceptual insight with convincing empirical performance and merits acceptance.